

**Estimation of Evapotranspiration and Other Soil Water Budget Components in an**
**Irrigated Agricultural Field of a Desert Oasis, Using Soil Moisture Measurements**
Zhongkai Li[a,b,d], Hu Liu[a,b*], Wenzhi Zhao[a,b], Qiyue Yang [a,b], Rong Yang [a,b], Jintao Liu[c]
*a Linze Inland River Basin Research Station, Chinese Ecosystem Research Network, Lanzhou 730000, China*
*b. Key Laboratory of Ecohydrology of Inland River Basin, Northwest Institute of Eco-Environment and Resources, Chinese Academy of Sciences, Lanzhou, 730000, China*
*c. State Key Laboratory of Hydrology-Water Resources and Hydraulic Engineering, Hohai University, Nanjing 210098, China*
*d. University of Chinese Academy of Sciences*
*\* The corresponding author, lhayz@lzb.ac.cn*
## Abstract
An accurate assessment of soil water budget components (*SWBCs*) is necessary for improving irrigation strategies and optimizing
the use of fertilizer in any water-limited environment such as the desert oases in arid northwestern China. However, quantitative
information of *SWBCs* is usually challenging to obtain, because, since the water cycle is principally driven by irrigation (*I*), drainage
(*D*), and evapotranspiration (*ET*) in desert oasis settings, none of the drivers can be easily measured under actual conditions. Soil
moisture is a variable that integrates the water balance components of land surface hydrology, and the evolution of soil moisture is
assumed to contain the memory of antecedent hydrologic fluxes, and thus can be used to determine *SWBCs* from a hydrologic
balance. A database of soil moisture measurements from six experimental plots in the middle Heihe River Basin of China (NT1 to
NT6, designed to investigate the long-term effects of cropping systems and agronomic manipulation on soil property evolution in
the ecotone of desert and oasis) was used to test the potential of a soil moisture database in estimating the *SWBCs*. The experimental
plots were treated as continuous pasture cropping, maize cropping, maize cropping with straw return, maize-maize-pasture rotation,
maize-pasture rotation, and maize-pasture intercropping. We first compared the hydrophysical properties of the soils in the plots,
including soil bulk density ($\rho_b$), vertical saturated hydraulic conductivity ($K_s$), and soil water retention features, and then determined
evapotranspiration and other *SWBCs* through a data-driven method that combined both the soil water balance method and the inverse
Richards function. Our results showed that although the tillage and planting of the past decade have significantly increased the soils'
water-holding ability, the magnitude of increase in most of the parameters was independent of the treatments applied across the
plots. Despite the relatively flat topography and consciously uniform irrigation, significant variances were observed among the plots
in both the cumulative irrigation volumes (between 652.1 mm at NT3 and 1186.5 mm at NT1) and deep drainages (between 170.7
mm at NT3 and 651.8 mm at NT1) during the growing season of 2016. Obvious correlation existed between the volume of irrigation
and that of drained water. However, the ET demands for all the plots behaved pretty much the same, with the cumulative ET values
ranging between 489.1 and 561.9 mm for the different treatments in 2016, suggesting that the irrigation amounts had limited
influence on the accumulated ET throughout the growing season. This work also confirmed that relatively reasonable estimations
of the *SWBCs* in a desert oasis environment can be derived by using soil moisture measurements, and the results will provide a great
potential for identifying appropriate irrigation amounts and frequencies, and thus move toward sustainable water resources
management, even under traditional surface irrigation conditions.
## Keywords
Evapotranspiration, Soil water budget, Desert oasis, Soil moisture, Inverse Richards Equation.
## 1. Introduction
Arid inland river basins in Northwestern China are unique ecosystems consisting of ice and snow, frozen soil, alpine vegetation,
oases, deserts, and riparian forest landscapes, in a delicate eco-hydrological balance (Liu *et al.*, 2015). Among these inland basins,
the Heihe river basin (HRB) is one of largest (Chen *et al.*, 2007). The oasis plains in the middle reaches of the HRB have become
an important source of grains, including the largest maize seed production center in China (Yang *et al.*, 2015). Crop water
requirements in this region are supplied mainly by irrigation from the river and from groundwater (Zhou *et al.*, 2017). According to
Wang *et al.* (2014), agriculture consumes 80 to 90% of the total water resources in the HRB, and has fundamentally altered the
regional hydrological processes and even resulted in eco-environmental deterioration (Zhao and Chang, 2014). Traditional irrigation
has low efficiency (i.e., a high leaching fraction) (Deng *et al.*, 2006; Li *et al.*, 2017) and the extensive fertilization practices have
given rise to higher levels of potential nitrate contamination in the groundwater, because water and pollutants percolate into the deep



sandy soils of the desert oasis, which have low water-holding capacities (Zhao and Chang, 2014). It is crucial to adopt a mechanism
that can preserve the role of irrigation in food security, yet with minimal consumption of the already scarce water, in order to increase
water productivity and conservation. Reducing water drainage and thus nitrate contamination in groundwater, saving water, and
increasing water and nitrogen use efficiency, are turning out to be important steps toward sustainable agriculture in this region (Hu
*et al.*, 2008)—steps that are being implemented by developing effective irrigation schedules (Su *et al.*, 2014).
Because allowing the soil to dry out too much may adversely affect the yield and quality of crops, while irrigating too early can lead
to wasted water, loss of fertilizer by leaching, increased operating costs and drainage problems, and sometimes decreased crop yield
or quality (Wright, 1971), an efficient irrigation scheduling program should aim to replenish the water deficit within the root zone
while minimizing leaching below this depth (Bourazanis *et al.*, 2015). Accordingly, an accurate assessment of soil water budget
components (*SWBCs*) is necessary for improving the irrigation management strategies in the oasis field. However, quantitative
information of *SWBCs* is usually challenging to obtain (Dejen, 2015). In desert oasis settings, the water cycle is principally driven
by irrigation (*I*), drainage (*D*), and evapotranspiration (*ET*). None of these drivers is easily measured in practice, however. For
example, not even the optimal irrigation amount can be determined accurately: the two most common methods of measuring
irrigation water—water meters or indirect methods—pose both economic and operational challenges to water managers, due to the
wide spatial distribution of small fields throughout rural areas (Folhes *et al.*, 2009). Measurement of deep percolation is also difficult,
and reliable data are rare in practice, and thus percolation is often calculated as a residual of the water balance (Bethune *et al.*, 2008;
Odofin *et al.*, 2012). ET is another source of uncertainty inherent in water budget estimations (Dolman and De Jeu, 2010), and its
estimation is only possible through the application of mathematical models, and is commonly calculated by relying on reference ET
(*ET$_0$*) or potential ET (*PET*) (Ibrom *et al.*, 2007; Suleiman and Hoogenboom, 2007; Allen *et al.*, 2011; Wang and Dickinson, 2012).
Soil moisture is a variable that integrates the water balance components of land surface hydrology (Rodriguez-Iturbe and Porporato,
2005), and over time it can be used to develop a record of antecedent hydrologic fluxes (Costa-Cabral *et al.*, 2008). Indeed, the
possibility of using changes in soil water content to estimate evaporation and other *SWBCs* has long been recognized (McGowan
and Williams, 1980) (Koksal *et al.*, 2017). Many studies, including Schelde *et al.* (2011) and Guderle and Hildebrandt (2015), have
shown that highly resolved soil moisture measurements contain a great deal of information that can be used to accurately determine
ET and sink term, based on hydrologic balance, when the appropriate approach is used. Rahgozar *et al.* (2012) and Shah *et al.* (2012)
extended these methodologies to determine other components of the water budget, such as lateral flow, infiltration, interception
capture, storage, surface runoff, and other fluxes. Time domain reflectometry (TDR) has been widely used in many irrigating regions,
including the desert oasis of the middle HRB, during the last decade (Liu *et al.*, 2015), for automated measurement of soil water
dynamics, because of its flexibility and accuracy (Schelde *et al.*, 2011). As one of the efforts in this region, intensive TDR
measurement of soil moisture was conducted in a long-term field experiment that was originally designed to test the accumulative
impacts of different cropping systems (i.e., maize and alfalfa) and agronomic manipulation (i.e., succession cropping, crop rotation,
row intercropping) on soil property evolution in the ecotone of desert and oasis. So far, however, no works have been published on
testing the potential of using a soil moisture database as a data-driving method in this region.
Based upon a soil moisture database, as mentioned above, this work aimed to 1) investigate the performance of using soil moisture
measurements to determine *ET* and other *SWBCs* in the croplands of desert oases; 2) estimate the long-term effects of cropping and
agronomic manipulation on field water balances by comparing the estimated *ET* and *SWBCs* of differently treated plots; and 3)
determine the potential for using soil moisture measurements to improve irrigation strategies in the desert oasis.

## 2. Materials and Methods

### 2.1 Study area

The study sites were located in the transition zone between the Badain Jaran Desert and the Zhangye Oasis in the middle HRB (Fig.
1). More specifically, they were in the Linze Inland River Basin Research Station of the Chinese Academy of Science (39º21'N,
100º17'E, altitude 1382m). This region has a temperate continental desert climate. The annual average temperature is about 7.6ºC,
and the lowest and highest temperatures are -27ºC and 39.1ºC for winter and summer, respectively. The annual average precipitation
is 117 mm and the mean potential evaporation is about 2,366 mm/a. The annual dryness index is 15.9. About 60% of the total
precipitation, with low rainfall intensity, is received during July–September, with only 3% occurring during winter. Northwest winds



prevail throughout the year, with intense sandstorm activities in spring. This region was part of a sandstorm-eroded area, and the
research site was converted into an artificial oasis during the 1970s. As a result, the soil types are dominated by sandy loam and
sandy soil, and characterized by coarse texture and rapid infiltration (Zhao *et al.*, 2010). The local dominant species are *Scotch Pine,*
*Gansu poplar, wheat,* and *maize* (Liu *et al.*, 2015), and sand-fixation plant species (planted since the 1970s), including *Haloxylon*
*ammodendron*, *Elaeagnus angustifolia*, *Tamarix ramosissima*, *Nitraria sphaerocarpa*, and annual herbaceous species such as *Bassia*
*dasyphylla*, *Halogeton arachnoideus*, *Suaeda glauca* and *Agriophyllum squarrosum*. The growing season of these plants and forages
usually starts in early April and normally continues through the month of September (DOY 94-288, Julian days >0 ºC).

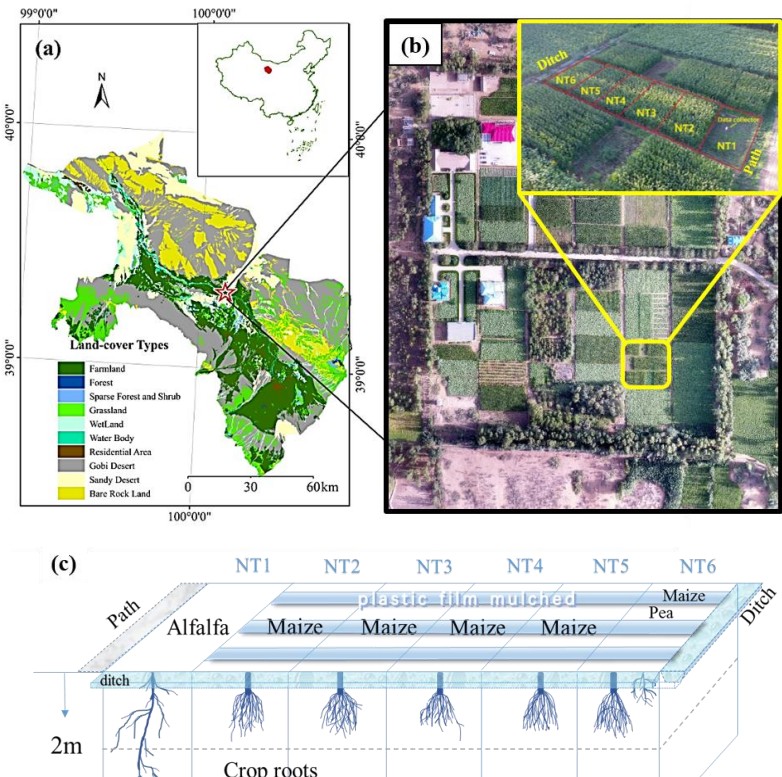



**Figure 1. a)** *Map of study area and research site;* **b)** *aerial view of the study site*; **c)** *detailed designs of the field experiments in 2016*
**2.2 Site description**
In order to investigate the accumulative effect of different cropping systems and agronomic manipulation on soil property evolution,
a long-term field experiment with six different treatments was set up in 2007. The experiment was performed with randomized
complete block design (RCBD) with three replications (Fig.1 b & c), so that in total, 18 plots of 6m × 9m were established. We
assumed that the soil texture and cultivation history (about 40 years) of the plots subjected to the different treatments were essentially
identical before the experiment was conducted. The middle one of the three replications (6 plots, NT1 to NT6) was selected for
installing the TDR sensors. The applied treatments of NT1 to NT6 were sequentially as follows: (1) continuous pasture cropping;
(2) continuous maize cropping; (3) continuous maize cropping with straw return; (4) maize-maize-pasture rotation; (5) maize-
pasture rotation; (6) maize-pasture intercropping. Plastic film mulching was applied during the initial growing season, and the
irrigation method was furrow irrigation (Zhao *et al.*, 2015). In 2016, NT1 was planted in alfalfa without plastic film mulch; NT2 to
NT5 in maize with plastic film mulch; and NT6 in interlaced maize (mulched) and peas (non-mulched) (Fig.1.c). Maize and peas
are annual crops, whereas alfalfa is a perennial forage legume which normally lives four to eight years; its root zone depth is between
1 and 2 m in the sandy soils of this region (Sun *et al.*, 2008). The growing season of maize and alfalfa in the region is usually from
early April till late September (Zhao and Zhao, 2014). Alfalfa was harvested twice during the growing season of 2016. Harvest 1




was conducted on 16 July, and the subsequent re-growth was harvested on 28 September (Su *et al.*, 2010).
The groundwater table depth fluctuated from 5 to 8 m at the experimental field during the year 2016. Irrigation with water extracted
from a nearby tube well was applied one by one in the plots from NT1 to NT6 during each irrigation event, and this work was
usually completed in 3 hours or less. The volumetric soil moisture of the six plots (NT1 to NT6) was measured with TDR systems
(5TE, Decagon Devices Inc. Pullman, WA, USA), which were installed at 5 different depths (20, 40, 60, 80, and 100 cm) at each
plot, with measurement intervals of 10 minutes. Before use, the TDR was calibrated from soil columns in the laboratory with known
volumetric water contents ($\theta_v$). A maximum likelihood fitting procedure was used to correct the observed data to eliminate the
potential errors induced by the soil texture and salinity (Muñoz-Carpena, 2009). Soil bulk density ($\rho_b$), vertical saturated hydraulic
conductivity ($K_s$), and soil water retention were determined using standard laboratory procedures on undisturbed soil cores in steel
cylinders (110 cm³ in volume, 5 cm in height) taken at 20-cm intervals down to 100 cm depth. Soil water retention curves were
measured at the pressure heads of -0.01, -0.05, -0.1, -0.2, -0.4, -0.6, -0.8, -1, -2, -5, -10, -15, -20, and -25 bars. $K_s$ was measured
with an undisturbed soil core using the constant head method (Salazar *et al.*, 2008). The values of field capacity ($\theta_{fc}$) and wilting
point ($\theta_w$) were empirically related to the corresponding soil water (matrix) potentials through the determined soil-water retention
curves (-0.1 bar for $\theta_{fc}$ and -15 bar for $\theta_w$). Hourly climatic data, including precipitation, temperature, radiation, wind, and
potential evaporation were recorded by a weather station located near the experimental site.
**2.3 Calculation methods**
**1) Water storage and irrigation amount**
Soil water storage ($S$) was calculated for the soil depth within the root zone (0-110 cm) based on the sensor readings through the
equation:

$$S = \sum_{i=1}^{5} \theta_i Z_i' \qquad (1)$$

where $\theta_i$ is the soil moisture of layer $i$; and $Z_i'$ is the layer thickness between 10cm above and 10cm below the sensor installation
depth. At the field level, examples of inflows are irrigation and rainfall, and examples of outflows are evaporation and deep leakage
beyond the root zone. An irrigation event usually lasted 20 to 30 minutes in each of the independent plots based upon the growth
stages of the plants. Soil moisture increased rapidly following irrigation events and decreased quickly as well during the subsequent
dry-down period. Rapid drying usually occurs for a few hours after a soil has been thoroughly wetted because of high water
conductivity (Fig. 2). The preferential flow was neglected in the selected soil profiles because the larger hydraulic conductivity of
sandy soil itself neutralizes the effects of preferential flow, and because coarse soil is relatively inimical to the formation of stable
preferential flow paths (Hamblin, 1985). Because the relatively short irrigation times that hampered the form of the steady infiltration
rate (Bautista and Wallender, 1993; Selle *et al.*, 2011), we hypothesized that no surface-water excess or steady-state flow took place
during any irrigation events, and assumed that deep percolation began when soil moisture storage reached maximum ($S_{max}$); thus
the irrigation volume ($V$) could be calculated as the difference between $S_{max}$ and $S_{ini}$:

$$V = S_{max} - S_{ini} \qquad (2)$$

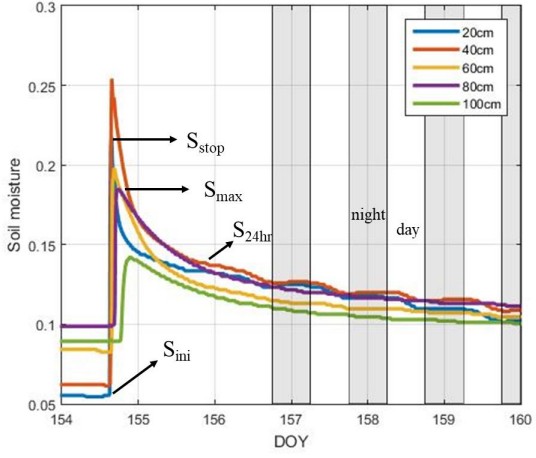






*Figure 2. Example diagram of the volumetric soil water content at various depths of NT6 during and after the irrigation event of 107.1 mm on DOY 154-160 (2016). $S_{stop}$: irrigation event ends, and moisture of uppermost soil layer starts to decrease; $S_{max}$: water storage maximum: after this point, deep percolation begins; $S_{24hr}$: deep percolation ends one day later; after this point, ET dominates the water-loss processes; $S_{ini}$: pre-irrigation, soil moisture minimum. The gray stripes between 156-160 DOY represent nights, i.e., 6:00 pm to 6:00 am of the next day.*

**2) Drainage and evapotranspiration**

Following irrigation water applications, the drainage behavior of soils consists of two stages: 1) rapid drainage and 2) slow drainage. During irrigation, the root zone becomes effectively saturated, and rapid drainage follows, leading to deep percolation. Then, as the water content in the soil falls, the hydraulic conductivity decreases sharply, as does the rate of drainage. The second phase, slow drainage, may continue for several days or months, depending on the soil texture (Bethune *et al.*, 2008). We assumed that rapid drying or drainage ceased 24 hours after an irrigation event, and thus rapid drainage ($Q_1$) could be estimated through the variances of water storage and actual ET during the period (Eq. 3). The actual ET during the period was assumed to be equal to the potential ET, because ET occurs unhindered with no water shortage.

$$Q1 = S_{max} - S_{24hr} - ET_p \qquad (3)$$

where $S_{24hr}$ is the soil moisture storage 24 hours after irrigation; $S_{max}$ is the maximum water storage after irrigation; and $ET_p$ is the potential ET during that day.

Slow drainage is especially important for sandy soils (Bethune *et al.*, 2008), as along with ET, it dominants the water loss processes during the second drying stage before the next irrigation event. Following Zuo *et al.* (2002) and Guderle and Hildebrandt (2015), an inverse method was employed to estimate the slow drainages and the average root water uptakes by solving the mixed theta-head formulation of the 1-D Richards Equation (Eq. 4) and iteratively searching for the sink term profile that produces the best fit between the numerical solution and the measured values of soil moisture content. ET is then obtained by summing rainfall and the sink term ($S_p$), and the drainage for this period is estimated as the water flux across the lower boundary of the soil profile. The above-mentioned 1-D Richards Equation is written as:

$$C(h)\frac{\partial h}{\partial t} = \frac{\partial}{\partial t}\left[K(h)\left(\frac{\partial h}{\partial z} - 1\right)\right] - Sp(z,t); \qquad (4)$$

$$h(z,0) = h_0(z) \qquad 0 \le z \le L; \qquad (5)$$

$$\left[-K(h)\left(\frac{\partial h}{\partial z} - 1\right)\right]_{z=0} = -E(t) \quad t > 0; \qquad (6)$$

$$h(L,t) = h_l(t) \qquad t > 0; \qquad (7)$$

where $h$ is the soil matric potential (cm); $C(h)$ the soil water capacity (cm$^{-1}$); $K(h)$ the soil hydraulic conductivity (cm d$^{-1}$); $h_0(z)$ the initial soil matric potential in the profile (cm); $E(t)$ the soil surface evaporation rate (cm) and $h_l(t)$ the matric potential at the lower boundary (cm); $L$ the simulating depth (cm); and $z$ the vertical coordinate originating from the soil surface and moving positively downward (cm). the iterative procedure runs the numerical model over a given time step ($\Delta t$) in order to estimate the soil water content profile $\tilde{\theta}_i^{v=0}$ at the end of the time step, assuming that the sink term $\widetilde{Sp}_{im,i}^{(v=0)}$ is zero over the entire profile at the beginning, where ~ depicts the estimated values at the respective soil layer $i$, and $v$ indicates the iteration step. Next, the sink term profile $\widetilde{Sp}_{im,i}^{(v=1)}$ is set equal to the difference between the previous approximation $\tilde{\theta}_i^{v=0}$ and the measurements $\theta_i$, while accounting for soil layer thickness and the length of the time step for units. In the following iterations, $\tilde{Sp}_{im,i}^{(v)}$ was used with the Richards equation to calculate the new soil water content $\tilde{\theta}_i^v$. The new average sink term $\widetilde{Sp}_{im,i}^{(v+1)}$ was then determined with Eq. (8):

$$\widetilde{Sp}_{im,i}^{(v+1)} = \widetilde{Sp}_{im,i}^{(v)} + \frac{\tilde{\theta}_i^v - \theta_i}{\Delta t} \cdot d_{z,i}; \qquad (8)$$

A backward Euler with a modified Picard iteration finite differencing solution scheme was adopted to inversely obtain the solution, and this implementation follows exactly the algorithm outlined by Celia *et al.* (1990). Three steps proposed by Guderle and Hildebrandt (2015), were taken to determine when the iteration process could be terminated in this calculation:

a.  Evaluate the difference between the estimated and measured soil water contents (Eq. 9) and compare the change in this difference to the difference from the previous iteration (Eq. 10):

$$e_i^{(v)} = \left|\theta_i - \tilde{\theta}_i^v\right| \qquad (9)$$
$$\varepsilon_{GH,i}^{(v)} = \left|e_i^{(v-1)} - e_i^{(v)}\right| \qquad (10)$$

b.  In soil layers where $\varepsilon_{GH}^{(v)} < 0$, set the root water uptake rate back to the value of the previous iteration $\widetilde{Sp}_{im,i}^{(v+1)} = \widetilde{Sp}_{im,i}^{(v-1)}$, since the current iteration was no improvement. Only if $\varepsilon_{GH}^{(v)} \ge 0$, go to the next step.





c.  If $e_i^{(v)} > 1 \times 10^4$, calculate $\widetilde{Sp}_{im,i}^{(v+1)}$ according Eq. (8); otherwise the current iteration sink term ($\widetilde{Sp}_{im,i}^{(v+1)} = \widetilde{Sp}_{im,i}^{(v)}$) is retained,

as it results in a good fit between estimated and measured soil water content.

**3) Boundary setting and data collection**
To reduce computational complexity, uniform soil profiles were assumed because there were no significant stratification differences
within the sandy soils (Table2) (Liu *et al.*, 2015). The upper boundary of the calculation was set as the atmospheric boundary
condition, and the calculation involved actual precipitation, irrigation, and potential evapotranspiration rates for the crop cover. The
surface fluxes were incorporated by using the average hourly rates distributed uniformly over each hour. The lower boundary
condition was set as a free drainage boundary because the groundwater table depth (deeper than 3.5m) was far below the crop
effective root depth during the growing season, and any capillary rise from groundwater could be ignored in this study. A unit vertical
hydraulic gradient boundary condition (i.e., $h = -5cm$) was implemented in the simulation in the form of a variable flux boundary
condition. The drainage rate $q(n)$ assigned to the bottom node $n$ was determined by the software as $q(n) = -K(h)$, where $h$ is the
local value of the pressure head and $K(h)$ is the hydraulic conductivity corresponding to this pressure head (Odofin *et al.*, 2012).
The meteorological measurements were monitored at the nearby weather station and were used to compute the upper boundary
condition. The potential ET used to force the boundary conditions was calculated with the Penman-Monteith combination equation
using hourly environmental data during the period from 1 April to 30 September (Fig. 3). We used soil moisture dynamics measured
in the soil profiles as inputs to inversely solve for sink term profiles at each plot for each hour (Lv, 2014). The soil moisture
measurements of 10-minute intervals during the period were hourly averaged to numerically filter out the noise associated with
highly resolved data. This had the effect of slightly reducing the infiltration and ET estimates, but this effect in the overall results is
negligible according to Guderle and Hildebrandt (2015). The actual amount of water delivered for irrigation ($Q_0$) was determined
from the power consumption of water pumping ($P_0$) through a relationship established between the power consumption and the
water pumping: $Q_0 = P_0 \times \eta$, where $\eta$ is the ratio of the power consumption per unit water pumped and is likely to be different
for different pumping heads. The coefficient was experimentally determined to be $8.5\ m^3 kW^{-1} h^{-1}$ for a head corresponding to
0.95 kg/cm$^2$ of delivery pressure in this study.

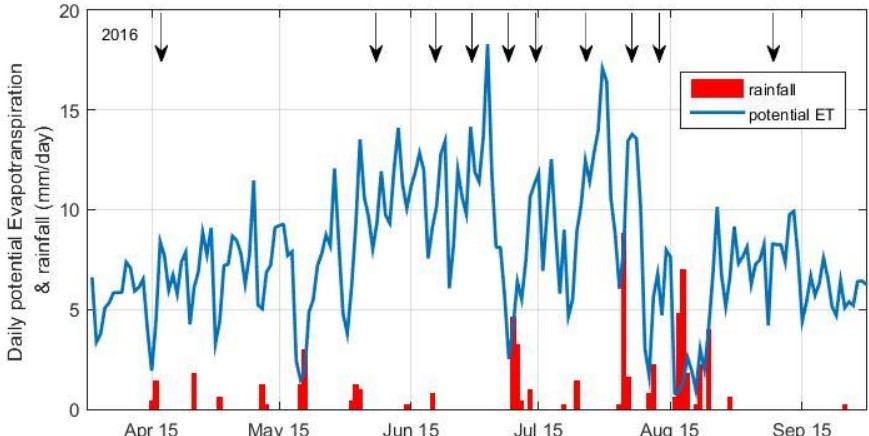


**Figure 3.** *Measured daily rainfall and potential ET estimated with the Penman-Monteith method during the growing season of 2016 at Linze*

*Station. The cumulative rainfall during the growing season was 69.2mm in 2016, and the black down arrows represent irrigation events.*


**Table 1.** *Nomenclatures involved in this study*

| V | irrigation amount for one irrigation event (mm) | K(h) | soil hydraulic conductivity (cm d$^{-1}$) |
|---|---|---|---|
| S | soil water storage (mm) | h$_0$(z) | initial soil matric potential in the profile (cm) |
| $S_{stop}$ | soil moisture storage when irrigation was stopped (mm) | E(t) | soil surface evaporation rate (cm) |
| $S_{ini}$ | soil moisture storage before irrigation start (mm) | h$_l$(t) | matric potential at the lower boundary (cm) |
| $S_{24hr}$ | soil moisture storage 24 hours after irrigation (mm) | L | simulation depth (cm) |
| $S_{max}$ | maximum soil water storage during irrigation event (mm) | z | vertical coordinate originating from the soil surface and moving positively downwards (cm) |
| $\theta_i$ | volumetric soil water content of layer $i$ (100%) | $\bar{\theta}_i^{y=0}$ | soil water content profile of soil layer $i$ at the beginning of each calculation |



| $\theta_v$ | theoretical volumetric water content calculated by the ratio of soil volume to water volume (100%) | $\widetilde{Sp}_{im,i}^{(v=0)}$ | sink term of soil layer $i$ at the beginning of irrigation, assuming it is zero |
|---|---|---|---|
| $\eta$ | ratio of the power consumption per unit water pumped | $d_{z,i}$ | thickness of soil layer $i$ |
| t | time | ~ | estimated values at soil layer $i$ |
| Q | steady-state drainage (mm) | $v$ | iteration step |
| $ET_p$ | potential ET during irrigation day (mm) | $\bar{\theta}_i^v$ | soil water content of step $v$ |
| $Z_i'$ | detection range of TDR, i.e., 20 cm | $\widetilde{Sp}_{im,i}^{(v)}$ | average sink term of step $v$ |
| $Sp$ | sink term, i.e., water extraction by roots, evaporation, etc. (cm) | $\Delta t$ | given time step |
| h | soil matric potential (cm) | $\varepsilon_{GH,i}^{(v)}$ | difference between and |
| C(h) | soil water capacity (cm$^{-1}$) | $e_i^{(v)}$ | difference between estimated and measured soil water content |
| $Q_0$ | real amount of water delivered for irrigation (m3) | $P_0$ | power consumption (kWh) |
| $D_{seas}$ | theoretical drainage volume over entire growing season in 2016 (mm) | $R_{seas}$ | cumulative rainfall during entire growing season in 2016 (mm) |
| $V_{seas}$ | theoretical irrigation volume over entire growing season in 2016 (mm) | $ET_{seas}$ | theoretical ET volume during entire growing season in 2016 (mm) |
| $\Delta S$ | difference in soil water storage before and after the growing season (mm) | $\rho_b$ | soil bulk density (g/cm$^3$) |
| $K_s$ | saturated water conductivity (cm/day) | $\theta_s$ | saturated water content (100%) |
| $\theta_{fc}$ | field capacity (100%) | $\theta_w$ | wilting point (100 %) |
| $S_w$ | wilting point (100 %) | $S^*$ | water stress point (100 %) |
| $S_{fc}$ | field capacity (100%) | $S_1$ | saturated water content (100%) |

## 3. Results

### 3.1 Soils' hydrophysical properties

A summary of most important hydrophysical characteristics of the soils at 0–100 cm depth (NT1 to NT6, and two other representative fields) in relation to their capacity for water storage is listed in Table 4. The textures were largely loamy sandy in the plots of NT1-NT6, in contrast to the sandy loam soil in an old oasis field with a long tillage history (~100 years) and the sandy soil in the desert with no tillage history (Table 2). Their bulk densities were generally between 1.4 and 1.5 g/cm$^3$—slightly higher than that in the local desert land, but still lower than that in maize fields of the old oasis. $\theta_s$, $\theta_{fc}$ and $\theta_w$ of the plots showed the same tendency of increasing soil hydrophysical properties (toward better water retention) as the bulk densities (Table 2). However, those parameters of the soil profiles are very similar to each other, especially between the same soil depths (horizontal) of the plots, suggesting that the different planting systems had similar influences on the soil hydrophysical proprieties, at least at the scale of 10 years. The effects of different cropping systems on soil moisture release characteristics are shown in Fig. 4. As expected, the relationship between soil water potential and volumetric water content across all data and treatment combinations followed a curvilinear pattern, where the water potential increased exponentially as soil water content increased.

The profile averaged values of saturated drainage velocity ($K_s$) were 119, 129.36, 286.04, 189.42, 207.92, and 216.14 cm day$^{-1}$ at NT1-NT6, respectively, which are coherent with the permeability results obtained in the laboratory with soil cores obtained from the same soils (Table 2). The large and varying values of $K_s$ showed a great drainage potential in the coarse-textured soil and an obvious heterogeneity in both horizontal and vertical profiles across the six plots. Soil moisture characteristic curves (SMC) in the six profiles are shown in Fig. 4, which indicates almost the same soil water content of NT1-NT6 under the same suction head, i.e., all the soil profiles were nearly saturated when the water potential reached the -0.01 bar and little was available after the soil water potential dropped to the -15 bar. Two obvious inflection points were observed, at $\theta \cong 0.08$ and 0.3, $\psi \cong -0.32$ and -15.2 bar in each of soil moisture characteristic curves from NT1-NT6. The slopes of the soil water potential-moisture, especially the parts between the inflection points of the six plots, were very close to each other, and also similar to that of the desert soil, suggesting similarly poor water capacities of the sandy soils (S *et al.*, 2002). A very significant difference in water capacities was observed when comparing the SMC of NT1-NT6 with that of the old oasis field, indicating that a considerably long period of time is still needed, for high soil water capacity to evolve, for these experimental sites.

*Table 2. Soil physical characteristics in the six experiment plots and two other selected plots around the study site*

| | NT1 | | | | | NT2 | | | | | NT3 | | | | | NT4 | | | | |
|---|---|---|---|---|---|---|---|---|---|---|---|---|---|---|---|---|---|---|---|---|
| | $K_s$ | $\rho_b$ | $\theta_s$ | $\theta_{fc}$ | $\theta_w$ | $K_s$ | $\rho_b$ | $\theta_s$ | $\theta_{fc}$ | $\theta_w$ | $K_s$ | $\rho_b$ | $\theta_s$ | $\theta_{fc}$ | $\theta_w$ | $K_s$ | $\rho_b$ | $\theta_s$ | $\theta_{fc}$ | $\theta_w$ |
| 20 cm | 47.2 | 1.38 | 0.36 | 0.25 | 0.09 | 183 | 1.46 | 0.34 | 0.19 | 0.08 | 44.3 | 1.40 | 0.36 | 0.21 | 0.09 | 54.1 | 1.39 | 0.38 | 0.21 | 0.08 |
| 40 cm | 46.8 | 1.55 | 0.33 | 0.21 | 0.06 | 82.1 | 1.55 | 0.32 | 0.15 | 0.05 | 259 | 1.54 | 0.34 | 0.18 | 0.06 | 266 | 1.50 | 0.36 | 0.17 | 0.06 |
| 60 cm | 166 | 1.48 | 0.35 | 0.20 | 0.06 | 118 | 1.53 | 0.34 | 0.20 | 0.05 | 73.8 | 1.53 | 0.35 | 0.19 | 0.05 | 355 | 1.47 | 0.36 | 0.16 | 0.06 |
| 80 cm | 61.0 | 1.45 | 0.33 | 0.17 | 0.05 | 164 | 1.48 | 0.35 | 0.18 | 0.05 | 1007 | 1.46 | 0.35 | 0.18 | 0.05 | 192 | 1.47 | 0.35 | 0.20 | 0.06 |
| 100 cm | 273 | 1.46 | 0.34 | 0.18 | 0.05 | 99.7 | 1.49 | 0.34 | 0.15 | 0.05 | 46.1 | 1.44 | 0.35 | 0.16 | 0.05 | 80.0 | 1.40 | 0.37 | 0.23 | 0.06 |
| $\bar{X}$ | 119 | 1.46 | 0.34 | 0.20 | 0.06 | 129 | 1.50 | 0.34 | 0.17 | 0.06 | 286 | 1.47 | 0.35 | 0.18 | 0.06 | 189 | 1.45 | 0.36 | 0.19 | 0.06 |




| | NT5 | | | | | NT6 | | | | | Maize field in old oasis | | | | | Local desert land | | | | |
|---|---|---|---|---|---|---|---|---|---|---|---|---|---|---|---|---|---|---|---|---|
| $SD$ | 99.6 | 0.06 | 0.01 | 0.03 | 0.02 | 42.8 | 0.04 | 0.01 | 0.02 | 0.01 | 413 | 0.06 | 0.01 | 0.02 | 0.02 | 126 | 0.05 | 0.01 | 0.03 | 0.01 |
| | $K_s$ | $\rho_b$ | $\theta_s$ | $\theta_{fc}$ | $\theta_w$ | $K_s$ | $\rho_b$ | $\theta_s$ | $\theta_{fc}$ | $\theta_w$ | $K_s$ | $\rho_b$ | $\theta_s$ | $\theta_{fc}$ | $\theta_w$ | $K_s$ | $\rho_b$ | $\theta_s$ | $\theta_{fc}$ | $\theta_w$ |
| 20 cm | 121 | 1.42 | 0.37 | 0.24 | 0.09 | 89.6 | 1.50 | 0.32 | 0.25 | 0.09 | 28.8 | 1.61 | 0.38 | 0.29 | 0.11 | 42.5 | 1.46 | 0.36 | 0.16 | 0.05 |
| 40 cm | 168 | 1.46 | 0.34 | 0.19 | 0.07 | 575 | 1.53 | 0.33 | 0.20 | 0.06 | 20.2 | 1.61 | 0.37 | 0.28 | 0.12 | 48.1 | 1.46 | 0.35 | 0.17 | 0.05 |
| 60 cm | 41.3 | 1.39 | 0.40 | 0.29 | 0.09 | 66.5 | 1.45 | 0.37 | 0.18 | 0.05 | 37.4 | 1.56 | 0.38 | 0.28 | 0.10 | 30.9 | 1.44 | 0.39 | 0.20 | 0.07 |
| 80 cm | 38.3 | 1.49 | 0.37 | 0.21 | 0.05 | 331 | 1.50 | 0.34 | 0.18 | 0.04 | 76.3 | 1.59 | 0.37 | 0.24 | 0.09 | 33.3 | 1.45 | 0.33 | 0.18 | 0.05 |
| 100 cm | 671 | 1.47 | 0.34 | 0.19 | 0.06 | 18.6 | 1.47 | 0.35 | 0.14 | 0.04 | 47.5 | 1.58 | 0.40 | 0.29 | 0.12 | 26.9 | 1.43 | 0.28 | 0.17 | 0.03 |
| | | | | | | | | | | | | | | | | | | | | |
| $\bar{X}$ | 208 | 1.45 | 0.36 | 0.22 | 0.07 | 216 | 1.49 | 0.34 | 0.19 | 0.06 | 42 | 1.59 | 0.38 | 0.28 | 0.11 | 36 | 1.45 | 0.34 | 0.17 | 0.05 |
| $SD$ | 265 | 0.04 | 0.02 | 0.04 | 0.02 | 234 | 0.03 | 0.02 | 0.04 | 0.02 | 22 | 0.02 | 0.01 | 0.02 | 0.01 | 9 | 0.01 | 0.04 | 0.02 | 0.01 |

$K_s$: *saturated water conductivity (cm/day); $\rho_b$: bulk density (g/cm3): $\theta_s$: saturated water content (100%); $\theta_{fc}$: field capacity (100%) and $\theta_w$:*
*wilting point (100 %); $\bar{X}$: mean value of the five soil layer; SD: Standard deviation of the five soil layer.*

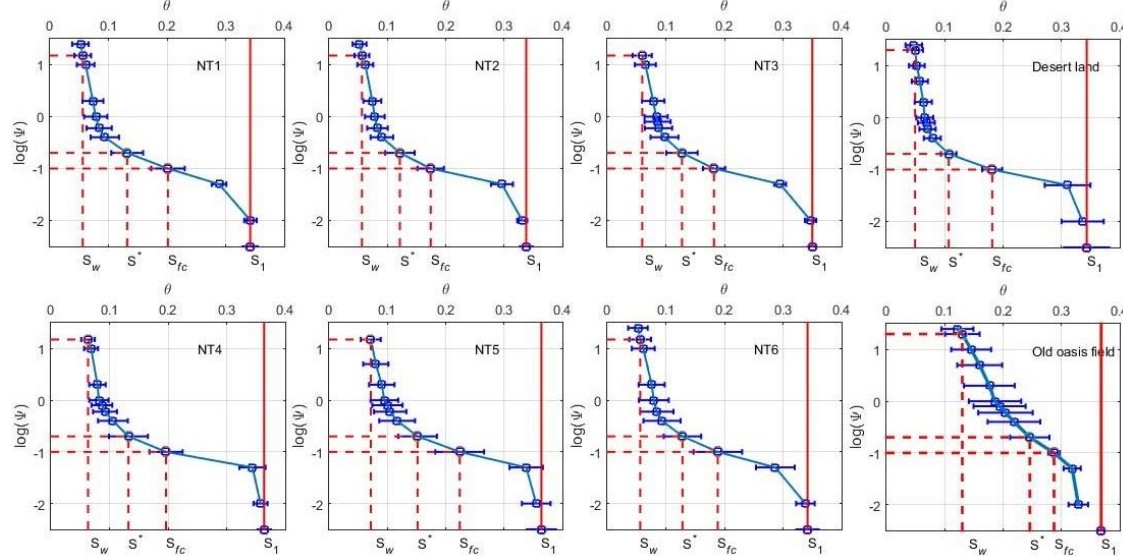

***Figure 4.*** *Soil moisture characteristic curve (SMC) of uniform soil profiles of the six experiment plots and two other representative fields. Soil*
*field capacity ($S_{fc}$), wilting point ($S_w$), and water stress point, i.e., point of incipient stomatal closure ($S^*$) are empirically related to the*
*corresponding soil matric potentials (-0.1 bar for $S_{fc}$, -0.2 bar for $S^*$ and -15 bar for $S_w$); the blue horizontal line represents the error bar, and*
*the solid red line represents saturated water content ($S_1$), which was obtained via the traditional Soil Drying method with 3 repetitions in each*
*layer; for soil water (matric) potential ($\Psi$) take the absolute value, for example, -0.01 bar is equal to -2 on the Y axis.*

## 3.2 Meteorological and irrigation data

The mean temperature of the growing season in 2016 was 27.12°C, or 3.12 degrees Celsius warmer than the long-term average of
the growing seasons in 2007-2016 (24.0°C), and the mean rainfall during the period was about 60.2 mm, or 47 percent less than the
long-term average of 115.4 mm (2005-2016), indicating that the weather was hotter and drier during the growing season in 2016
than in the previous ten years. Irrigation was delivered at a rate of 2250 L·ha$^{-1}$·min$^{-1}$ by way of traditional furrow irrigation. Fig. 7
presents a summary of the amount of water applied over the entire growing season of 2016. Irrigation applications began in mid-
April and continued until late September, every 5 to 25 days, depending upon moisture content and crop growth (Fig. 3). A total of
10 irrigation events were sequentially applied through furrow irrigation for the plot during the entire growing season. The cumulative
irrigation volumes for the plots of NT1 to NT6 during the period were about 1187, 760, 652, 840, 683, and 867 mm, respectively.
The estimated average irrigation crop demand within the plots was 831.6 mm, which compares well with the actual irrigation
volume (868.8 mm) determined through power consumption, suggesting that the calculated irrigation agrees closely with the
measured values from the farm fields when accurate irrigation and rainfall data are available. A difference of 4.5% in the irrigation
amount was observed between the real values and the measured values over the entire growing season of 2016, indicating a high
reliability of the water balance method used in *SWBCs* estimation.

## 3.3 Soil moisture dynamics (SMDs)

Fig 2 shows an example of the soil water content responses at various depths of NT6 during and after the irrigation event of 107.1
mm on DOY 154 (2016). TDR measurements exhibited a sharp increase when irrigation began and then decreased rapidly as it was
turned off, due to the poor water-holding capacity of the sandy soil. The increase in water content occurred layer by layer from the




upper horizons, suggesting limited influence from potential preferential flow (Liu and Lin, 2015), while the rapid moistening of the
deep horizons could imply the existence of water loss by drainage. The greatest rate decrease in water content was observed in the
top 20 cm of soil. During the 12 h after irrigation, the water content at the top sensor decreased from 21.9% to 14.2%. For the same
interval of time the water contents in 40, 60, 80 and 100 cm depths of soil decreased from 25.4%, 19.8%, 18.5% and 14.2% to
15.7%, 14.3%, 15.4% and 12.8%, respectively. After irrigation ended, water continued to move down the soil profile; and thus the
top part of the profile was continuously losing water to the soil below it. The lower soil horizons were leaching water into the
horizon below but at the same time were receiving water that had drained from the horizon immediately above, resulting in lower
rates of decrease in water content for these layers than for those at the top horizon (20 cm) (Fares and Alva, 2000). Very similar
patterns of changes in water content through the six different soil profiles were observed.
The average field capacity value ($\theta_{fc}$) of NT6 determined from laboratory measurement of soil water release curves was 19%.
Within 24 hours after the end of irrigation, the soil moisture values for the all the measured horizons (20-100 cm depth) of NT6
ranged between 12.3% and 14.2%, lower than the field capacity (Fig.2), suggesting that the rapid drainage of water away from the
root zone soil (0-100 cm) was terminated during the period, as expected. In the mornings of the subsequent days, the decrease in
soil moisture again sped up as the evaporative demand of the atmosphere gradually increased. In the absence of any irrigation during
the subsequent nights, a slow-down or even a very light increase in the soil moisture content was observed in the top soil layer (Fig
2). We checked all the soil moisture time series of NT1-NT6 during the entire growing season period (Fig.5), and no constant water
content throughout the entire soil profile was detected in any of those selected plots, suggesting that our previous hypothesis that no
steady-state flow took place during any irrigation events was supported. According to the data, there was also no obvious response
of soil moisture regimes to precipitation, indicating a very limited contribution of rainfall to the soil water storage compared with
irrigation. In fact, more than 90% of the rainfall events in this region are less than 5 mm (Fig. 3), and canopy interception (about 2-
5 mm) and strong potential evaporation may have hampered any effective infiltration from those precipitation events.

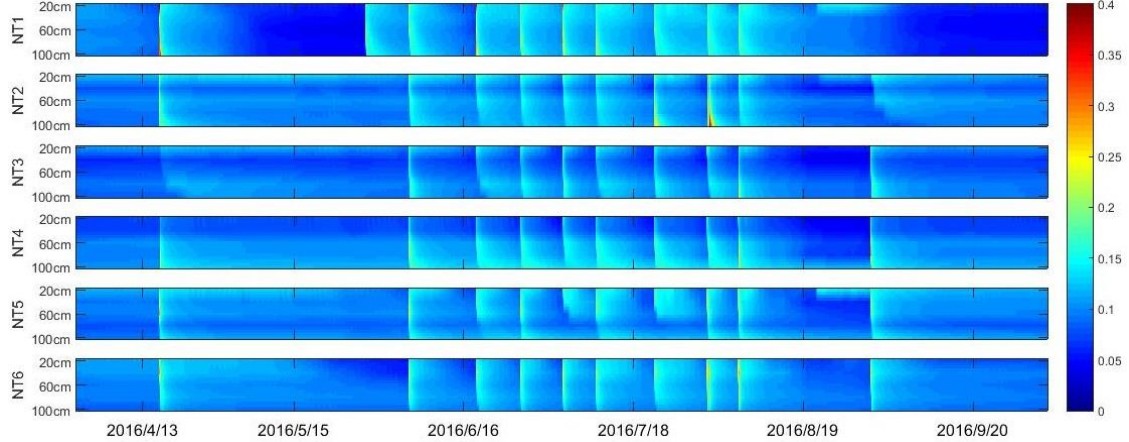

***Figure 5.****Spatial and temporal variations of soil water content with a time resolution of ten minutes. The color bar on the right side represents*
*volumetric soil water content. Time period was from Apr.1 to Oct.1, 2016. Irrigation events for NT2-6 occurred on 4/16, 6/2, 6/15, 6/23, 7/1, 7/7,*
*7/18, 7/28, 8/3, and 8/28. NT1 had one more irrigation event on 5/25 and one less on 8/28.*

### 3.4 Soil water budget components (*SWBCs*)

The estimated soil water budget components, including total irrigation, deep percolation, and *ET*, at the six different plots during
the growing season of 2016 are summarized in Table 3 and Fig. 7. Evapotranspiration and deep percolation dominated the fields'
relatively simple soil water budgets during the study period. A clear trend in seasonal variation of the water budget components can
be observed at the site (Fig. 7). The corresponding ET values were very similar for all the plots. Three different stages of ET could
be discriminated throughout the 2016 growing season: ET rate was very low at the initial stage (i.e., the first 50 days of the growing
season), and increased gradually as LAI became greater with crop development, before reaching maximal values at the mid-season
stage. After that, ET decreased gradually until harvest time. The estimated daily ET values ranged largely between 0.2 and 12 mm
d$^{-1}$, with an average of 3 mm d$^{-1}$. No significant differences were detected in the daily ET when Duncan's multiple range test was



applied at the 5% level to compare among the six experimental plots (*P*>0.75). A relatively large difference was observed between selected plots in this study, i.e., significant higher cumulative irrigation volume was found at NT1. The relative facility with which an excess of water in the soil was produced caused an important deep percolation, which became greater as it progressed further up the irrigation gradient. Among the plots, 45-79% of the input irrigation water was consumed by way of ET (i.e. for plant growth), while the change in soil water storage before and after the growing season was quite small. It is clear that although there was a high correlation between the volume of irrigation and that of drained water, the irrigation amount had limited influence on the accumulated ET during the growing season.

*Table 3. Estimated evapotranspiration and other major soil water budget components during the growing season of 2016*

| Cumulative SWBCs | NT1 | NT2 | NT3 | NT4 | NT5 | NT6 |
|---|---|---|---|---|---|---|
| Irrigation | 1186.5 | 760.1 | 652.2 | 840.4 | 683.2 | 867.3 |
| Drainage | 651.8 | 288.3 | 170.7 | 340.1 | 212.4 | 364.7 |
| ET | 534.6 | 489.1 | 508.8 | 561.9 | 539.2 | 538.1 |
| Storage diff.* | -52.7 | 0.17 | 3.6 | 2.2 | 5.44 | -11.64 |

*\* Storage differences represent the difference in soil water storage before and after the growing season.*

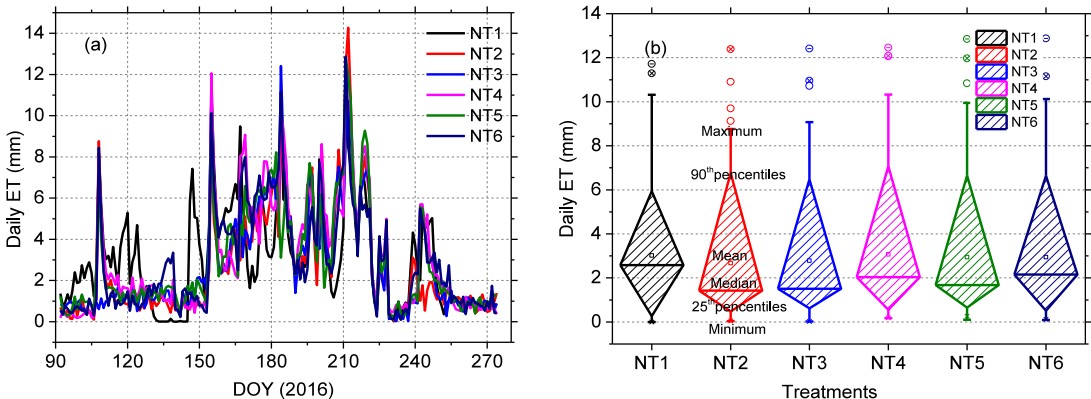

*Figure 6. Daily ET during the growing season of 2016 as determined from the inverse Richards method: a) time series of estimated daily ET, b) box-and-whisker diagrams showing the minimum, median, 25th percentile, 75th percentile, and maximum daily ET. No significant differences were detected when Duncan's multiple range test was applied at the 5% level to compare values among the plots.*

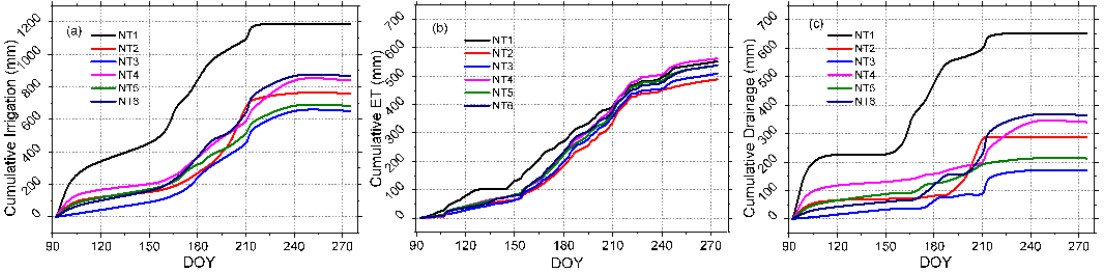

*Figure 7. Estimated water components of the plots during the growing season of 2016: a) cumulative irrigation, b) cumulative ET, c) cumulative drainage*

## 4. Discussion

### 4.1 Estimated ET

Cumulative ET values calculated from inverse Richards methods ranged between 489.1 and 561.9 mm for the different treatments in 2016. The values of ET obtained from the current study are well within the range of published ET values at the nearby sites (406-778 mm), and are consistent with the averages from other studies (~585.5mm) also done in this region, including Zhao and Ji (2010); Rong (2012); Yang *et al.* (2015); You *et al.* (2015); Zhao *et al.* (2015), etc. for maize fields similar to the ones present at the study site (Table 4). Compared with the methods used in the literatures listed in Table 4, the soil moisture data-driven method used in this





paper is more reliable because it produced the best fit between the numerical solution and the measured values of soil moisture
content, even with vertical flow accounted for (Guderle and Hildebrandt, 2015). The narrow range of cumulative ET (489.1-561.9
mm) observed in 2016 can be attributed to the similar sandy soil texture and mesic moisture regimes caused by frequent irrigation
(Fig. 5), which in turn suggested that both cropping systems and agronomic manipulation had limited influence on the accumulated
ET during the growing season (Srivastava et al., 2017). This result is well supported by the evidence reported by early investigators,
that the ET differences in different cropping systems are smaller for coarse-textured soils compared with fine-textured soils (Jalota
and Arora, 2002), and that ET is strictly a function of ambient atmospheric conditions under normal or wet conditions (Rahgozar et
al., 2012).
The observed seasonal trend of ET corresponded well to the irrigation frequency and crop water consumption characteristics of the
growth stage (Fig. 7), and similar patterns in the ET processes have also been reported by many other works conducted in this region
(Zhao et al., 2010; Zhao et al., 2015). Although we also noticed that the cumulative ET of NT1 was relatively higher than those of
the other plots at the beginning of growing season, this phenomenon can be largely attributed to the plastic film mulching at the
other five plots. In the early growing season (seeding to emergence), soil evaporation (E) is the major part of ET (Zhao et al., 2015),
and the plastic film mulching applied to NT2 to NT6 was able to significantly retain the soil moisture and thus decrease soil
evaporation (Jia et al., 2006). However, the differences in the cumulative ET, between NT1 and the other plots, were quite small
after the mid-growing season, most likely because with the plant canopy development, crop transpiration became the major portion
of ET, and the influence of plastic film on ET diminished (Jia et al., 2006; Qin et al., 2014; Zhang et al., 2017). Another influence
that may have decreased the evapotranspiration at NT1 after the mid-growing season is cutting. Cutting alfalfa lowers the leaf area
index (LAI) and drastically changes the effective diffusive resistance, consequently lowering the daily ET rate of alfalfa at NT1,
although for a short time after cutting, evaporation from the soil surface may compensate for the decrease in transpiration (Dong et
al., 2003; Su et al., 2010).
**Table 4.** *Reported ET of oasis maize field in the middle Heihe River Basin (HRB)*

| ET (mm) | Growing period | Year | Soil type | Irrigation | Rainfall | Methods | Paper |
|---|---|---|---|---|---|---|---|
| 651.6 | Apr.11-Sep.18 | 2001 | --- | 690 | 84.4 | Water balance methods | (Su et al., 2002) |
| 513.2 | Apr.16-Sep.22 | 2005 | Light loam | 360 | 153.5 | Bowen ratio method | (Jinkui et al., 2007) |
| 486.2 | Apr.16-Sep.22 | 2005 | Light loam | 360 | 153.5 | Reference ET-crop coefficient method | (Jinkui et al., 2007) |
| 777.75 | Apr.21-Sep.15 | 2007 | Sandy loam | 1194 | 102.1 | Bowen ratio method | (Zhao et al., 2010) |
| 693.13 | Apr.21-Sep.15 | 2007 | Sandy loam | 1194 | 102.1 | Penman | (Zhao et al., 2010) |
| 618.34 | Apr.21-Sep.15 | 2007 | Sandy loam | 1194 | 102.1 | Penman-Monteith | (Zhao et al., 2010) |
| 615.67 | Apr.21-Sep.15 | 2007 | Sandy loam | 1194 | 102.1 | Water balance method | (Zhao et al., 2010) |
| 560.31 | Apr.21-Sep.15 | 2007 | Sandy loam | 1194 | 102.1 | Priestley-Taylor | (Zhao et al., 2010) |
| 552.07 | Apr.21-Sep.15 | 2007 | Sandy loam | 1194 | 102.1 | Hargreaves method | (Zhao et al., 2010) |
| 671.2 | Apr.10-Sep.20 | 2009 | Sandy loam | 797 | 97.7 | FAO-56-PM and dual crop coefficient method | (Zhao and Ji, 2010) |
| 640 | Apr.10-Sep.20 | 2009 | --- | 797 | 97.7 | Shuttleworth-Wallace dual-source model | (Zhao et al., 2015) |
| 570—607 | Apr.22-Sep.23 | 2010 | Loamy sand | 990-1103 | 75 | Field experiments | (Rong, 2012) |
| 405.5 | Apr.20-Sep.22 | 2012 | Clay loam | 553 | 95.9 | Water balance and isotope methods | (Yang et al., 2015) |
| 450.7 | Apr.20-Sep.22 | 2012 | --- | 430 | 104.9 | Eddy covariance system | (You et al., 2015) |
| 554.0 | Apr.20-Sep.22 | 2012 | --- | 430 | 104.9 | Penman | (You et al., 2015) |
| 489-562 | Apr.10-Sep.20 | 2016 | Sandy soil | 652-867 | 60.2 | Inverse method | This paper |


## 365 4.2 Other estimated *SWBCs*

The irrigation volume of maize (NT2 to NT6) within our plots ranged between 652.2 and 867.3 mm, with an average value of 760.6
mm, which is well comparable to the range of average maize field irrigation volume in this region, i.e., a range between 604.8 and
811.4 mm reported in the Statistical Yearbook of Zhangye City for the period of 1995 to 2017 (see http://www.zhangye.gov.cn).
When compared to the other treatments of plastic film mulching, significantly higher amounts of the applied irrigation (1186.5 mm)
were found in NT1, which could be attributed to the larger percentage of infiltrating surface area and the relatively longer irrigation
duration caused by rougher surface of the ground without plastic film mulching. According to Yang et al. (2018), plastic film mulch
has been widely used to increase the productivity of crops in arid or semiarid regions of China. The logic behind this approach is
that plastic film mulch improves the soil physical properties, such as the soil water content and temperature in the top soil layers,
and thus leads to increased plant growth and yield (N. Mbah et al., 2010). Our results suggested that plastic film mulching can
equally reduce irrigation duration and applied water depth by lowering surface roughness and thus the friction coefficient of the
ground. Similar results were also reported by earlier investigators (Jia et al., 2006; Qin et al., 2014; Zhang et al., 2017).
A less extreme but still significant difference can be found in the irrigation volumes (~652.2 to 867.3 mm) over the other five plots
with plastic film mulching (NT2-6). This may be associated with the inconsistent durations caused by uneven irrigation applications,



randomly rough soil surfaces, and mutation of the infiltration rate (i.e., $K_s$) across the plots (Table 2). Uneven irrigation may be
further attributed to the uneven fields and ditches, which may lead to the application of much more water than required for
evapotranspiration, in some places (Babcock and Blackmer, 1992). Soil surface texture has a direct effect on soil water and complex
interactions with other environmental factors (Yong et al., 2014). The hydraulic behavior and the rate of traditional surface irrigation
is eventually influenced by the inflow and duration of each irrigation (Ascough and Kiker, 2002). Although only slight differences
exist among the retention curves (Fig. 4), the differences in saturation water conductivity ($K_s$) can be substantial (varying between
119 cm/day at NT1 and 286 cm/day at NT3), indicating that a slight difference in hydrophysical properties of soil profiles could be
amplified to generate wildly varying infiltration behavior, especially during saturated or near-saturated stages under actual irrigation
conditions (Ojha et al., 2017).
Estimated deep drainage rates were observed, ranging from 170.7 mm (NT3) to 651.8 mm (NT1), amounting to about 26.2% and
54.9% of the total irrigation of the two plots, respectively. Compared with the theoretical deep drainage determined by water balance
techniques (Rice et al., 1986), an error of -2.6 to 43.1 mm, or 0.2 % to 17.6%, was obtained for the cumulative deep drainage (Table
3), indicating the reliability of the method used to estimated deep drainage in this study. Drainage within the maize fields ranged
from 170.7 mm to 364.7 mm, which are in good agreement with other results from the same region, i.e., 255 mm through isotopes
obtained by Yang et al. (2015), and 339.5 mm through the Hydrus-1D model by Dong-Sheng et al. (2015). The data expressed in
Fig. 2 also explain how easily an excess of water, and therefore deep drainage, can occur in these soils. Indeed, the deep drainage
was directly proportional to the amount of irrigation applied during any particular period (Fig. 7, Table 3). This phenomenon is easy
to understand because for a given amount of irrigation, the likelihood of a drainage event and its average size both increased naturally
with the irrigation amount (Fig.7) (Keller, 2005). It is obvious that drainage should be an essential part of irrigation design and
management. According to our results, an average of 40.6% of input water was consumed by deep leakage across the six plots; this
is unproductive and could even cause nutrient loss and groundwater pollution at field scales (Fares and Alva, 2000), suggesting
there is a huge potential for increasing irrigation water-use efficiencies and reducing irrigation water requirements in this region.
**4.3 Long-term effects on soil water budgets**
Long-term cropping can increase annual water productivity by improving soil hydrophysical properties and reducing unproductive
water losses (Caviglia et al., 2013). Through the physical mechanical actions and active release of chemicals, crop root systems may
create heterogeneity in soil properties (Hirobe et al., 2001; Read et al., 2003); this and other similar feedbacks between long-term
planted crops and the soil environment change water flow and soil hydraulic characteristics, and thus affect local water balances
(Baldocchi et al., 2004; Séré et al., 2012). Although it is difficult to quantify the consequences of plant-soil feedbacks on the
hydrologic cycle of farmland, because of the lack of an accurate simulation model (Jalota and Arora, 2002), our results indicated
that the tillage and planting of past decades have significantly increased the soil water holding ability (i.e., higher values of $\rho_b$, $\Theta_s$,
$\Theta_{fc}$ and $\Theta_w$ compared with the sandier land). The magnitude of increase in most of the parameters, except $K_s$ in soil vertical
profiles, was independent of the treatments applied across the six selected plots, which also suggests that different cropping systems
and agronomic manipulation have limited effects on differing soil physical characteristics in sandy soil, at least at a decade scale,
and this agrees well with the reports from Katsvairo et al. (2002). However, we argue that significant differences in soil
hydrophysical properties among the plots may occur if the treatments are conducted over longer periods of time, i.e., ~100 years or
more.
**4.4 Potential for *SWBC* estimation by using soil moisture measurements**
Information on SWBCs is crucial for irrigation planning at both the field and regional scale (Jalota and Arora, 2002), and the best
estimates should be based on models of soil water, because direct measurements are not available in most cases (Campbell and Diaz,
1988). Many studies including modeling work have been conducted in this region during the past decades (Table 4). Since there has
been a lack of accurate parameters to assess the heterogeneity and complexity involved in modeling (Ibrom et al., 2007; Suleiman
and Hoogenboom, 2007; Allen et al., 2011; Wang and Dickinson, 2012), however, most of these were rough approximations based
on meteorological methods and water balance equations (Ji et al., 2007; Rong, 2012; Wu et al., 2015; Yang et al., 2015; Jiang et al.,
2016). Data-driven methods have been considered one of the most promising ways to directly determine ET and other SWBCs (Li
et al., 2002; Guderle and Hildebrandt, 2015), and many possible options, including single- or multi-step, and single- or multi-layer
water balance methods, have been proposed and tested with synthetic time series of water content (Guderle and Hildebrandt, 2015).
Our results suggest that a combination of a soil water balance method and the inverse method could be a good candidate for SWBC





estimation in this region, and can provide a reliable solution, especially in regards to estimating ET, root water uptake, and water vertical flow, and do not require any prior information of root distribution parameters, while they can be applicable under both wet and dry weather conditions (Guderle and Hildebrandt, 2015).

Early researches suggested that decreasing the irrigation amount and increasing the irrigation frequency is the best choice for saving water and improving water use efficiency in the middle HRB (Ji *et al.*, 2007; Rong, 2012; Wu *et al.*, 2015; Yang *et al.*, 2015; Jiang *et al.*, 2016). This scenario can be achieved not only by adopting proper modern irrigation systems but also by integrating new technologies into the effective planning of irrigation schedules, so that plants can be supplied with optimal water volume and minimum water loss. Soil water budget models help in translating irrigation amounts in different time periods to evapotranspiration (ET), which has significance from the standpoint of crop yield (Jalota and Arora, 2002). Our results show that superfluous irrigation has no effect on increasing ET, because of the poor water-holding capacity of the sandy soil in this region, and thus irrigation application should not exceed a specific threshold (i.e., root zone depletion, ~527 mm for maize) to avoid deep percolation, which has a negative effect: increasing irrigation costs (Zotarelli *et al.*, 2016). However, water deficits in crops and the resulting water stress on plants also influences crop evapotranspiration and crop yield (Kallitsari *et al.*, 2011). Thus, a soil moisture measurement method based on SWBC estimation makes it possible to quantify water budget components for different time periods, and has great potential to identify appropriate irrigation amounts and frequencies, thus moving toward sustainable water resources management, even under traditional surface irrigation conditions (Tawara *et al.*, 2015).

## 5. Conclusions

A database of soil moisture measurements in 2016 from a long-term field experiment (which was originally designed to test the accumulative impacts of different cropping systems and agronomic manipulation on soil-property evolution in the ecotone of desert and oasis) conducted in the middle Heihe River Basin of China was used to test the potential of a soil-moisture time series in estimating the *SWBCs*. We compared the hydrophysical properties of the soils in the plots, and then determined evapotranspiration and other *SWBCs* through a data-driven method that combines both the soil water balance method and the inverse Richards function. Our results showed that although the tillage and planting of the past decade have significantly increased the soil water-holding ability, the magnitude of increase in most of the parameters was independent of the treatments applied across the plots, at least during a 10-year period. Despite the relatively flat topography and similar soil hydrophysical properties, significant variances were observed among the plots in both cumulative irrigation volumes (between 652.1 mm at NT3 and 1186.5 mm at NT1) and deep drainages (between 170.7 mm at NT3 and 651.8 mm at NT1) during the growing season of 2016. Obvious correlation existed between the volume of irrigation and that of drained water. However, the ET demands for all the plots behaved pretty much the same, with the cumulative ET values ranging between 489.1 and 561.9 mm for the different treatments in 2016, suggesting that superfluous irrigation has no effect on increasing ET because of the poor water-holding capacity of the sandy soil in this region. This work confirmed that a relatively reasonable estimation of the *SWBCs* in a desert oasis environment can be derived through a data-driven method using soil moisture measurements, and the estimated results of the *SWBCs* will provide a great potential for optimizing irrigation strategies, thus moving toward sustainable water resources management in this water-limited environment.

## Acknowledgements

This research was supported in part by the National Natural Science Foundation of China (Grant 91425302, PI: Shaozhong Kang and Wenzhi Zhao) and the Youth Innovation Promotion Association of Chinese Academy of Sciences (Grant, PI: Hu Liu).

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
