# Peer review of "Hydrology and § Earth System Sciences Discussions"

_Hydrology and Earth System Sciences, 2018_

## Short Comment (SC1) · 17 Oct 2018

**Short Comments**

I read the paper by Li et al. with great interest as we are doing similar research activities, i.e., the inversion of the soil water balance equation to obtain estimates of rainfall (e.g., *Brocca et al., 2014*) and irrigation (*Brocca et al., 2018*; *Zaussinger et al., 2018*) through soil moisture observations (no need to cite them in the paper, it's just for in-

formation). In our research, we mainly use satellite observations and therefore we are testing the capabilities to estimate rainfall and irrigation through satellite soil moisture products, assumed not accurate.

In the paper by Li et al. it is mostly assumed that in situ soil moisture observations are perfect and through the inversion of the soil water balance equation accurate estimates of water fluxes, i.e., irrigation, evapotranspiration and drainage, are obtained. While I mostly agree that in situ soil moisture data can be used for obtaining water fluxes, i.e., the soil water budget components, I also believe that uncertainties in the inversion approach are present.

The main issue I found is exactly related to the uncertainties in the inversion procedure. For instance, the six soil moisture probes show significant differences in the amount of irrigation for different probes; irrigation at NT1 is nearly double (1-8 times) of NT3. I do not understand how it might be possible as I believe that the amount of irrigation applied in the field is the same for all probes. Am I wrong? Indeed, the irrigation estimates strongly depend on the assumption behind equation (2) and, specifically, in the estimation of Smax. Why is the comparison with in situ observations of irrigation not done? I believe it is needed definitely.

Anyhow, I congrats the paper for the very interesting paper.

**References (not exhaustive)**

Brocca, L., Ciabatta, L., Massari, C., Moramarco, T., Hahn, S., Hasenauer, S., Kidd, R., Dorigo, W., Wagner, W., Levizzani, V. (2014). Soil as a natural rain gauge: estimating global rainfall from satellite soil moisture data. Journal of Geophysical Research, 119(9), 5128-5141, doi:10.1002/2014JD021489.

Brocca, L., Tarpanelli, A., Filippucci, P., Dorigo, W., Zaussinger, F., Gruber, A., Fernández-Prieto, D. (2018). How much water is used for irrigation? A new approach exploiting coarse resolution satellite soil moisture products. International Journal of Applied Earth Observation and Geoinformation, 73C, 752-766, doi:10.1016/j.jag.2018.08.023

Zaussinger, F., Dorigo, W., Gruber, A., Tarpanelli, A., Filippucci, P., Brocca, L. (2018). Estimating irrigation water use over the contiguous United States by combining satellite and reanalysis soil moisture data. Hydrology and Earth System Sciences Discussion, in review, doi:10.5194/hess-2018-388.

---

## Short Comment (SC2) · 31 Oct 2018

General Comments: Accurate assessment of soil water budget components (SWBCs) is necessary for improving irrigation strategies and optimizing the use of fertilizer in agricultural systems. However, quantitative information of SWBCs is usually challenging to obtain. Soil moisture is a variable that integrates the water balance components of land surface hydrology, and thus over time it can be used to develop a record of antecedent hydrologic fluxes. This paper presents an interest and important study on a soil moisture data-driven method for the water budget components estimation. Overall,

the research was well conducted, and the whole manuscript was generally well written. I recommend publication consideration, granted that the minor concerns and questions below are properly addressed.

Specific comments: - The abstract is concise and almost complete. The only suggestion is to clarify in line 27-28 that why the cumulative irrigation volumes varied so much among the 6 different plots, i.e., 652-1186mm. Although you mentioned that this phenomenon can be largely attributed to the plastic film mulching at the NT1 in your manuscript (Line 351-353), it should also be stressed in your abstract. -Page 2, Line 53-54, "…irrigating too early", I guess it should be "irrigating too much"? -Page 2, Line 69-70, the citation of the references [(Mcgowan and Williams, 1980) (Koksal et al., 2017)] was not correct due likely to font conversion. -Page 3, Fig. 1c, the root systems of crop and alfalfa drawn by the authors are not good to indicate their real patterns in the soil profiles. In most cases, root distribution with depth is that of a negative exponential function, i.e., (Wasson et al., 2017). -Page 5, Line 165-194, there are more than one method are available to do this calculation, i.e., enhanced soil water balance, slope approach (Guderle and Hildebrandt, 2015), what is the reason for choosing the Inverse Soil Water Flow Model in this paper? -Page 6, Line 206, what kind of software? You mean the codes that you developed to do this calculation? You don't mention it throughout the manuscript. Please clarify this point. -Page 7, Line 246, the citation is not complete, please double check it. -Page 9, Line 287, why you start this part from NT6 rather than NT1, what is the logic behind this? -Page 11, Line 378-381, you mentioned that one of the potential reasons that could result in the different irrigation rates is the mutation of the infiltration rate. If this is the case, how well do the TDR measurements-based estimation of irrigations represent the experimental plots? More related discussions are needed to clarify this point. -Page 12, Line 416-417, "Information on SWBCs is crucial for irrigation planning at both the field and regional scale (Jalota and Arora, 2002), and the best estimates should be based on models of soil water, because direct measurements are not available in most cases", it is better to move this sentence to the 2nd paragraph in section 4.4., i.e., Line 429. -Page 15, Line 542;

Page 17, Line 619, the references were not organized in the correct style of HESS.

---

## Referee Comment (RC1) · J. Niu (Referee) · 13 Nov 2018

Estimation of soil water components is crucial for the optimum irrigation strategies, especially for the ecologically fragile region in Northwest China. The soil moisture is regarded as a more integrated one as it may have the memory features for the antecedent hydrological effects. This study tried to estimate the fields-scale water budget based on six experimental plots with different cropping patterns, NT1 to NT6. The evapotranspiration and soil water components are calculated by employing data-driven method, which combines the soil water balance method and the inverse Richards function. The manuscript is generally well-structured and the study is well-founded. There are several concerns/suggestions that may be helpful for the improvement listed as below.

1. The research findings should be more effectively highlighted in the Abstract. There are some too detailed information listed in the current abstract.

2. How to explain the largest Ks value 1007 for NT3 in Table2?

3. Where is the rainfall observation station?

4. Wrong explanation for $\varepsilon$ in Table1. There are wo same explanations with "field capacity" and "wilting point" in Table 1, but no explanation for soil water potential $\Psi$ in Table 1.

5. For the case of NT1, both the irrigation amount and drainage amount are the highest, but no big differences about ET with other cases. Based on the energy balance, this high amount of irrigation may reduce the temperature profile (see Chen et al. 2018, Soil of the Total Environment, for the irrigation effects on energy balance in the Heihe River basin), which possibly affect actual ET requirement for the crops in NT1.

6. Following the above point, the inverse Richards method employed for the soil water budget in this study did not consider the energy effects of different irrigation scenarios, which is possible to affect the water budget during different phases. This should be discussed.

7. Another concern is the scale issue. As we have recognized the larger variability for different sets, we can imagine how difficult for a large agricultural land. So the potential for using soil moisture measurement to improve irrigation strategies is still on the way.

8. Line 443: I don't think one-year experiment could be called as "long-term" for hydrological processes.

9. There are many duplicate sentences between Abstract and Conclusions.

---

## Referee Comment (RC2) · Anonymous Referee #2 · 14 Nov 2018

ET is very important in water resource management. This manuscript presented an experimental study combing with soil water balance modeling and mathematical modeling. The research is worthy, well structured and written. But I do not recommend to publish it in present form. The main reasons include; (1ïïjĽThe irrigation amount is the most important component in this study and needs to accurately measured. Unfortunately the manuscript did not provide reliable information on it. 1) two irrigation amounts were used in this study, one calculated from the difference between Smax and Sini, the other estimated from power consumption. But I do understand which is

the real value and which is the measured value (L272). The definition of Smax was not clearly given. 2) according to the text, the irrigation was delivered at a rate of 2250L/ha/min (L264) and each irrigation event lasted 20-30 min, therefore the irrigation amount was about 6 mm for each time and total irrigation amount was about 60 mm for the growing season. If it is not mistake, it is much lower than the estimation and is not acceptable. 3) I think it is possible to make accurate measures with water meter in such small experimental plot. (2) About the mathematical model, the setting is quite important. In section 2.3, 2), equation (7) shows that the lower boundary was defined by soil matric potential (L175), which in section 2.3 3), it was set to free drainage (L202). It is not acceptable. Moreover, root uptake is also an important factor in water redistribution simulation, but the manuscript did not give the information about the root distributions of the crops. (3) About the location of the TDR systems in the experimental plots. Because the irrigation method is furrow irrigation, the soil water contents are different with location. From table 2, the saturate water content is about 34% and the field capacity is about 20%. If the preferential flow is limited, as suggested in the manuscript, the water content should be higher than 20%, even 24h after the irrigation event. But the measured value is only 21.9% for Smax and 14% for S24. Therefore I doubt the TDR systems were not in the suitable location. Therefore the information is not sufficient and correct.
* * *

---

## Referee Comment (RC3) · Anonymous Referee #3 · 16 Nov 2018

General comments:

Arguments of this paper need to be stated more clearly. Sub-sections 3.1, 3.2 and 3.3 should be moved to the Section 2. Conclusion section repeats the results to significant extent, while responding to the arguments very weakly. The novel contributions of this work should be presented explicitly. Readers may feel confused by finding out the focusing points: the improvement of methodology or new scientific findings?

Specific comments:

[Figure]

1-2: Estimation of Evapotranspiration and Other Soil Water Budget Components in an Irrigated Agricultural Field of a Desert Oasis, Using Soil Moisture Measurements

Comment: (1)evapotranspiration is one of the soil water balance components. Is it necessary to let it stand out here? (2)what are the key issues to be addressed in this paper. A clear definition to the problem is needed.

14-15: water cycle is principally driven by irrigation (I), drainage (D), and evapotranspiration (ET) in desert oasis settings

Comment: Water cycle is primarily driven by evaporation demand under influence of irrigation. Soil water percolation may occur when too much water applied to the root zone. Anyway, it is not proper to say that water cycle is driven by irrigation and drainage.

24-25: ... through a data-driven method that combined both the soil water balance method and the inverse Richards function.

Comments: (1)It is not very common to say 'Richards function'. Instead, Richards equation is the most popular description. (2) data-driven? According to the manuscript, it is a soil-moisture data based method. This method is not uncommon.

31-32: ..., suggesting that the irrigation amounts had limited influence on the accumulated ET throughout the growing season.

Comment: With regard to this study, sufficient water was applied to each treatment and caused significant percolation, indicating that crops grew under non-water stress condition. However, it cannot be concluded generally that irrigation amount had limited influence on the accumulated ET. Otherwise, this may mislead both understanding and practice.

45: Traditional irrigation

Comment: What is the traditional irrigation? It should be defined specifically because it is different from place to place around the world.

[Figure]

58-50: In desert oasis settings, the water cycle is principally driven by irrigation (I), drainage (D), and evapotranspiration (ET). None of these drivers is easily measured in practice, however.

Comment: It is not proper to call all these components drivers of water cycle.

65-66:. . ., and its estimation is only possible through the application of mathematical models, and is commonly calculated by relying on reference ET(ET0) or potential ET (PET)

Comment: "only possible"? You might have not said it.

79-80: . . .oasis. So far, however, no works have been published on testing the potential of using a soil moisture database as a data-driving method in this region.

Comment: As prerequisite condition, it should not be locally limited. Otherwise, the value of the research could be discounted.

161: With no water shortage

Comment: It is better to phrase it as under non-water stress (condition)

164: The potential ET during that day

Comment: How is the potential ET calculated here? Reference ET, potential ET, maximum ET are different concepts.

199-200: The upper boundary of the calculation was set as the atmospheric boundary condition, and the calculation involved actual precipitation, irrigation, and potential evapotranspiration rates for the crop cover.

Commment: (1) how is the film mulching effects considered for the upper boundary condition? (2) how is the bare soil evaporation estimated as the upper boundary condition? (3) how is the upper boundary condition defined for the inter-cropping treatment? And the alternative mulching strips?

226: In Table 4

Commment: Table 2?

237: The profile averaged values of saturated drainage velocity ($\eth\check{I}\check{R}\xi\eth\acute{I}\check{S}\breve{a}$) were 119, 129.36, 286.04, 189.42, 207.92, and 216.14 cm day-1 at. . .

Comment: It is not necessary to list this values in the text because they are already given in the table.

269: . . .irrigation crop demand. . .

Comment: Irrigation demand, crop demand are meaningful concepts in crop water requirement studies. What does the irrigation crop demand mean?

292: . . ., a slow-down or even a very light increase. . ..

Comment: A slow down decrease or even light increase?

293-295: We checked all the soil moisture time series of NT1-NT6 during the entire growing season period (Fig.5), and no constant water content throughout the entire soil profile was detected in any of those selected plots, suggesting that our previous hypothesis that no steady-state flow took place during any irrigation events was supported.

Comment: What is the purpose of this sentence? For any frequently irrigated soil profile, it is hard to reach a steady flow state.

298: . . .and strong potential evaporation may have hampered any effective infiltration from those precipitation events.

Comment: It is the insufficient precipitation that attributes to the negligible infiltration rather than the strong evaporative demand.

310: . . .and increased gradually as LAI became greater with crop development, . . .

Comment: LAI has never been mentioned previously in the paper although it is very

important information supporting discussions in the later sections.

315-316: The relative facility with which an excess of water in the soil was produced caused an important deep percolation, which became greater as it progressed further up the irrigation gradient.

Comment: This sentence should be rephrased. It is confusing.

340: . . ., the soil moisture data-driven method. . .

Comment: The soil moisture data based method, might be a better description to this work.

341: . . .the best. . .

Comment: "the best" among which and which?

344-345: . . ., which in turn suggested that both cropping systems and agronomic manipulation had limited influence on the accumulated ET during the growing season, . . .

Conmment: This is correct when preconditioned only.

365: 4.2 Other estimated SWBCs

Comment: Does it mean the other SWBCs in this study or the SWBCs given by other people in the literatures?

401: 4.3 Long-term effects on soil water budgets

Comment: Does this manuscript involve any long-term issues, either the parameters or the water balance budgets?

---

## Author Comment (AC1) · 14 Dec 2018

**Response to Prof. Y. Shen (SC2)**

**General comments**

Accurate assessment of soil water budget components (SWBCs) is necessary for improving irrigation strategies and optimizing the use of fertilizer in agricultural systems. However, quantitative information of SWBCs is usually challenging to obtain. Soil moisture is a variable that integrates the water balance components of land surface hydrology, and thus over time it can be used to develop a record of antecedent hydrologic fluxes. This paper presents an interest and important study on a soil moisture data-driven method for the water budget components estimation. Overall, the research was well conducted, and the whole manuscript was generally well written. I recommend publication consideration, granted that the minor concerns and questions below are properly addressed.

**Response**: We thank Prof. Shen for taking the time to review our manuscript and for his generally positive feedback on our study.

**Specific comments**

1) The abstract is concise and almost complete. The only suggestion is to clarify in line 27-28 that why the cumulative irrigation volumes varied so much among the 6 different plots, i.e., 652-1186mm. Although you mentioned that this phenomenon can be largely attributed to the plastic film mulching at the NT1 in your manuscript (Line 351-353), it should also be stressed in your abstract.

**Response:** Thanks for pointing this out. We will add and change in the abstract the following sentences for clarification: "Despite the relatively flat topography and consciously uniform irrigation, significant variances were observed among the film-mulched plots (NT2-6) in both the cumulative irrigation volumes (between 652.1 mm at NT3 and 867.3 mm at NT6) and deep drainages (between 170.7 mm at NT3 and 364.7 mm at NT6) during the growing season of 2016. Moreover, the unmulched plots (NT1) has remarkable higher values either in cumulative irrigation volumes (1186.5 mm) or in deep drainages (651.8 mm) compare with other plots."

2) Page 2, Line53-54, ". . .irrigating too early", I guess it should be "irrigating too much"?

**Response: Yes**, you are right, that was a careless typo. We apologize for this, and we will correct it as suggested in the revision.

3) Page 2, Line 69-70, the citation of the references [(Mcgowan and Williams, 1980) (Koksal et al., 2017)] was not correct due likely to font conversion.

**Response:** This citation will be corrected in the revision.

4) Page 3, Fig. 1c, the root systems of crop and alfalfa drawn by the authors are not good to indicate their real patterns in the soil profiles. In most cases, root distribution with depth is that of a negative exponential function, i.e., (Wasson et al., 2017).

**Response:** Thanks for the nice suggestion. We will reorganize this figure in the revision as suggested.

5) Page 5, Line 165-194, there are more than one method are available to do this calculation, i.e., enhanced soil water balance, slope approach (Guderle and Hildebrandt, 2015), what is the reason for choosing the Inverse Soil Water Flow Model in this paper?

**Response:** Very good question, and yes, there were more than one method available to do this calculation. According to Guderle and Hildebrandt (2015), the regression method based on diurnal fluctuation of soil water contents (M1) and the inverse method based on solving Richards equation (M2) were proven as the two most accuracy and reliable methods among the four tested ones. However, we found that although M1 performs well in the synthetic data generated with soil water flow model, it cannot be easily used in practice because it does not consider the hysteresis effect of soil moisture between different soil layers, which in turn can result in abnormal values of ET. Therefore, we finally chose the inverse model to estimate the evapotranspiration and slow drainage. We will clarify this point in our revision.

6) Page 6, Line 206, what kind of software? You mean the codes that you developed to do this calculation? You don't mention it throughout the manuscript. Please clarify this point.
**Response:** Yes, we did mean the codes that we developed to do this calculation. It seems this description does not make sense in the manuscript, so that we will remove the statement of software, and change it in the revision as "The drainage rate $q(n)$ assigned to the bottom node $n$ was determined by the relationship as $q(n) = -K(h)$, where $h$ is the local value of the pressure head and $K(h)$ is the hydraulic conductivity corresponding to this pressure head (Odofin *et al.*, 2012)."

7) Page 7, Line 246, the citation is not complete, please double check it.
**Response:** We will correct this in the coming revision.

8) Page 9, Line 287, why you start this part from NT6 rather than NT1, what is the logic behind this?
**Response:** Sorry for the confusion that brings to you and to the readers. To clarify the issue, we reorganized this part as: "The average field capacity value ($\theta_{fc}$) of NT1-6 determined from laboratory measurement of soil water release curves was 19.2% (i.e., 20%, 17%, 18%, 19%, 22% and 19% for NT1-6 respectively). After 24 hours of the end of irrigation (June 3, 2016), the soil moisture values for the all the measured horizons (20-100 cm depth) of NT1-6 ranged between 8.9% and 16.9% (13.7-15.7%, 13.7-15.1%, 8.9-14.5%, 9.6-16.9%, 11.7-15.3% and 12.3-14.2% for NT1-6 respectively), lower than the field capacity (Fig.2&5), suggesting that the rapid drainage of water away from the root zone soil (0-100 cm) was terminated during the period, as expected."

9) Page 11, Line 378-381, you mentioned that one of the potential reasons that could result in the different irrigation rates is the mutation of the infiltration rate. If this is the case, how well do the TDR measurements-based estimation of irrigations represent the experimental plots? More related discussions are needed to clarify this point.
**Response:** Very constructive suggestion. Although the estimated average irrigation amount of the six experimental plots is well consistent with the actual average irrigation amount, we must agree that considerable uncertainties exist in the estimate of the irrigations. According to this suggestion and the comments from other reviewers, a new section (**4.5 Uncertainty analysis**) will be included in the revision to solve this concern.

10) Page 12, Line 416-417, "Information on SWBCs is crucial for irrigation planning at both the field and regional scale (Jalota and Arora, 2002), and the best estimates should be based on models of soil water, because direct measurements are not available in most cases", it is better to move this

sentence to the 2nd paragraph in section 4.4., i.e., Line 429.

**Response:** Thanks, we will move this statement into the 2nd paragraph of section (4.4) in the revision.

11) Page 15, Line 542; Page 17, Line 619, the references were not organized in the correct style of HESS.

**Response:** We will reorganize the references in the correct style of HESS.

**References:**

Guderle M, Hildebrandt A. 2015. Using measured soil water contents to estimate evapotranspiration and root water uptake profiles - a comparative study. Hydrology & Earth System Sciences, 19: 409-425.

---

## Author Comment (AC2) · 28 Dec 2018

**Response to Dr. J. Niu (RC1)**

**General comments**

Estimation of soil water components is crucial for the optimum irrigation strategies, especially for the ecologically fragile region in Northwest China. The soil moisture is regarded as a more integrated one as it may have the memory features for the antecedent hydrological effects. This study tried to estimate the fields-scale water budget based on six experimental plots with different cropping patterns, NT1 to NT6. The evapotranspiration and soil water components are calculated by employing data-driven method, which combines the soil water balance method and the inverse Richards function. The manuscript is generally well-structured and the study is well-founded. There are several concerns/suggestions that may be helpful for the improvement listed as below.

**Response**: We warmly thank the Dr. J. Niu for the overall favorable impression of the work, and for the constructive suggestions, with which our manuscript is significantly improved in both its clarity and organization. Please see our point-by-point response (in blue) in detail below.

**Specific comments**

1) The research findings should be more effectively highlighted in the Abstract. There is some too detailed information listed in the current abstract.

**Response**: These too detailed information listed in the abstract has been largely simplified, and only the most important ones will be kept in the revised version.

2) How to explain the largest $K_s$ value 1007 for NT3 in Table 2?

**Response**: As we all know, hydraulic conductivity can exponential change with soil water content in near-saturated or saturated conditions (Horton, 1992), so that the slight variation in soil physical property could potentially cause extreme value in $K_s$ (i.e., 1007 cm/day). Previous works done in this region have also shown that the value of $K_s$ in sandy soil could range between ~100 cm/day and >1000 cm/day in the sandy soils of the ecotone between desert and oasis (Yao *et al.*, 2013; Xiu-Rong *et al.*, 2014; Sun *et al.*, 2015). Thus, it is reasonable that such extreme value exists in the measured values of $K_s$. To eliminate the potential disruptions caused by extreme value of $K_s$, the averaged value of $K_s$ were used to do the inverse estimation for each soil profile. We will further clarify this point in our revision.

3) Where is the rainfall observation station?

**Response:** the weather station is located about 150 m from the experimental site. This information will be included in the revision.

4) Wrong explanation for ε in Table1. There are two same explanations with "field capacity" and "wilting point" in Table 1, but no explanation for soil water potential Ψ in Table 1.

**Response**: Thanks for pointing out the mismatch for the parameters in Table 1. It will be corrected in the revision.

5) For the case of NT1, both the irrigation amount and drainage amount are the highest, but no big differences about ET with other cases. Based on the energy balance, this high amount of irrigation may reduce the temperature profile (see Chen et al. 2018, Soil of the Total Environment, for the

irrigation effects on energy balance in the Heihe River basin), which possibly affect actual ET requirement for the crops in NT1. Following the above point, the inverse Richards method employed for the soil water budget in this study did not consider the energy effects of different irrigation scenarios, which is possible to affect the water budget during different phases. This should be discussed.

**Response:** Very useful suggestion. Yes, the high amount of irrigation may reduce the temperature of soil profile, because the irrigation usually accompanied by an increase of latent heat flux, which is often relate to evapotranspiration (Haddeland *et al.*, 2006; Zou *et al.*, 2017; Chen *et al.*, 2018). Due to the higher amount of irrigation, it is natural that NT1 should get higher ET than the other cases, however, no big differences were detected in ET values from our calculation. One reason behind this phenomenon is that all the six experimental plots were fully irrigated at each irrigation event, and most of the over-applied water at NT1 and occasionally occurred at other plots (NT2-6) were deep drained during the short periods following irrigation because of the poor water holding capacity of sandy soils (as evidenced in Figure 5), so that the differences in the temperature regimes in the soil profiles caused by the different irrigation volumes across the plots could be largely ignored, and thus ET is strictly a function of ambient atmospheric conditions (Rahgozar et al., 2012). We will add related discussion in the coming revision to solve this concern.

6) Another concern is the scale issue. As we have recognized the larger variability for different sets, we can imagine how difficult for a large agricultural land. So, the potential for using soil moisture measurement to improve irrigation strategies is still on the way.

**Response**: Yes, each soil moisture probe can monitor only a small volume in heterogeneous soils. We agree that the potential for using soil moisture measurement to improve irrigation strategies is still on the way. However, it still provided a valuable reference for coarse-textured soils like sandy soil in this region for improve the irrigation efficiency. Discussion upon this issue will be included in the revision.

7) Line 443: I don't think one-year experiment could be called as "long-term" for hydrological processes.

**Response:** OK, we will remove the word of "long-term" here to clarify this, although what we mean here is that the cropping experiment rather than the hydrological processes monitoring is a long-term work.

8) There are many duplicate sentences between Abstract and Conclusions.

**Response:** Following this suggestion, we will reorganize Abstract to make it more concise and remove the duplicated part with conclusions.

**References:**

Chen Y, Niu J, Kang S, Zhang X. 2018. Effects of irrigation on water and energy balances in the Heihe River basin using VIC model under different irrigation scenarios. Science of The Total Environment, **645**: 1183-1193.

Haddeland I, Lettenmaier DP, Skaugen T. 2006. Effects of irrigation on the water and energy balances of the Colorado and Mekong river basins. Journal of Hydrology, **324**: 210-223.

Horton R. 1992. Soil Physics. **21**.

Rahgozar M, Shah N, Ross MA. 2012. Estimation of evapotranspiration and water budget components using concurrent soil moisture and water table monitoring. International Scholarly Research Notices, **2012**: 1-15.

Sun L, Liu T, Duan L, Jia K. 2015. Prediction of saturated hydraulic conductivity of surface soil in sand dune and meadow interlaced region of horqin with pedo-transfer functions method. Acta Pedologica Sinica.

Xiu-Rong WU, Zhang FB, Wang ZL. 2014. Variation of sand and loess properties of binary structure profile in Hilly Region covered by sand of the Loess Plateau. Journal of Soil & Water Conservation.

Yao S, Zhao, Zhang. 2013. A comparison of soil saturated hydraulic conductivity (kfs) in different horqin sand land. Acta Pedologica Sinica.

Zou M, Niu J, Kang S, Li X, Lu H. 2017. The contribution of human agricultural activities to increasing evapotranspiration is significantly greater than climate change effect over Heihe agricultural region. Scientific Reports, **7**.

---

## Author Comment (AC3) · 28 Dec 2018

**Response to the anonymous reviewer 3# (RC3)**

We would like to thank the reviewer 3# for his accurate and frank review and used their precious suggestions to improve the paper. We tried to answer to all the comments made and we are ready to prepare and submit a new version of the manuscript. The point by point answers are written in blue.

**General comments**

Arguments of this paper need to be stated more clearly. Sub-sections 3.1, 3.2 and 3.3 should be moved to the Section 2. Conclusion section repeats the results to significant extent, while responding to the arguments very weakly. The novel contributions of this work should be presented explicitly. Readers may feel confused by finding out the focusing points: the improvement of methodology or new scientific findings?

**Response**: Thanks for the nice suggestion, according to which we reorganized the proposed questions in our work, and make it more focused on the improvement and implementation of the methodology. The novel contributions of this work also have been further clarified according the reviewer's suggestion. The reviewer also suggests us to move the Sub-sections 3.1, 3.2 and 3.3 to the Section 2. We spend much time to think about it, and finally decide to keep them at the original places, because the information presented in sub-sections 3.1, 3.2 and 3.3 are the results from our laboratory experiments and field observations, rather than a simply background introduction.

**Specific comments**

1) Line 1-2: Estimation of Evapotranspiration and Other Soil Water Budget Components in an Irrigated Agricultural Field of a Desert Oasis, Using Soil Moisture Measurements, **Comment:** (1) evapotranspiration is one of the soil water balance components. Is it necessary to let it stand out here? (2) what are the key issues to be addressed in this paper. A clear definition to the problem is needed.

**Response:** (1) Yes, we do think it is necessary to let evapotranspiration (ET) stand out, because ET is the most important one among all the soil water balance components (SWBCs), and the one the related researchers are most interested in, because of its direct relevance to the crop yield, and the fact that maximizing crop yield is the major objective of agricultural irrigation strategies (Kang *et al.*, 2002; Liu *et al.*, 2002; Zhang *et al.*, 2004). (2) The key issue we concerned in this paper is the potentials of soil moisture measurements in determining ET and other SWBCs in the croplands of desert oases environments.

2) Line 14-15: water cycle is principally driven by irrigation (I), drainage (D), and evapotranspiration (ET) in desert oasis settings, **Comment:** Water cycle is primarily driven by evaporation demand under influence of irrigation. Soil water percolation may occur when too much water applied to the root zone. Anyway, it is not proper to say that water cycle is driven by irrigation and drainage.

**Response:** Thanks for the nice suggestion, we have changed "water cycle" as "hydrological process of farmland", and cited this comment in the revision.

3) Line 24-25: through a data-driven method that combined both the soil water balance method and

the inverse Richards function. **Comments:** (1) It is not very common to say 'Richards function'. Instead, Richards equation is the most popular description. (2) data-driven? According to the manuscript, it is a soil-moisture data-based method. This method is not uncommon.

**Response:** (1) Thanks for the useful information, and the description of "Richards function" has been replaced with the more popular one ("Richards equation") in the revision; (2) As the reviewer suggested, "soil moisture data-based method" was adopted in the revision to replace the "data-driven method". We agree that the idea of "soil moisture data-based method" is not uncommon in literatures, because soil moisture measurements were used to estimate the infiltration by numerical solutions as early as 1950s (Gardner and Mayhugh, 1958; Hanks and Bowers, 1962). However, ET estimates with the inverse methods are recent developments, i.e., Zuo *et al.* (2002), Ross (2003) and Guderle and Hildebrandt (2015). Indeed, according to our knowledge and based on the literature search, only very few researches applied this method especially in arid environments and coarse-texture soils due to the limited availability of highly resolved soil moisture measurements, so that here we would argue this method is still novel, and it deserves more attention in future researches on agricultural water management. Our work investigated for the first time the performance of using soil moisture measurements to determine ET and other *SWBCs* in the croplands of desert oases. The estimated results of the *SWBCs* will provide a great potential for optimizing irrigation strategies, thus moving toward sustainable water resources management in water-limited environment.

4) Line 31-32: "suggesting that the irrigation amounts had limited influence on the accumulated ET throughout the growing season", **Comment:** Regarding this study, enough water was applied to each treatment and caused significant percolation, indicating that crops grew under non-water stress condition. However, it cannot be concluded generally that irrigation amount had limited influence on the accumulated ET. Otherwise, this may mislead both understanding and practice.

**Response:** Thanks for pointing this out. We have reorganized this statement in the coming revision as "suggesting that the superfluous irrigation amounts had limited influence on the accumulated ET throughout the growing season because of the poor water-holding capacity of the sandy soil".

5) Line 45: Traditional irrigation, **Comment:** What is the traditional irrigation? It should be defined specifically because it is different from place to place around the world.

**Response:** The traditional irrigation in this work was defined as flood irrigation, and it has been further clarified in the revision.

6) Line 58-50: In desert oasis settings, the water cycle is principally driven by irrigation (I), drainage (D), and evapotranspiration (ET). None of these drivers is easily measured in practice. **Comment:** It is not proper to call all these components drivers of water cycle.

**Response:** This sentence has been re-worded as follows: "In desert oasis settings, the hydrological process of cropland is principally driven by irrigation ($I$), drainage ($D$), and evapotranspiration ($ET$)".

7) Line 65-66: "its estimation is only possible through the application of mathematical models, and is commonly calculated by relying on reference ET(ET0) or potential ET (PET)", **Comment:** "only possible"? You might have not said it.

**Response:** This sentence has been re-worded as follows: "its estimation in field scale is usually through the application of mathematical models, and is commonly calculated by relying on reference ET

($ET_0$) or potential ET ($PET$)".

8) Line 79-80: . . .oasis. So far, however, no works have been published on testing the potential of using a soil moisture database as a data-driving method in this region. **Comment:** As prerequisite condition, it should not be locally limited. Otherwise, the value of the research could be discounted.

**Response:** This part will be re-organized in the coming revision to solve the concern.

9) Line 161: With no water shortage, **Comment:** It is better to phrase it as under non-water stress (condition).

**Response:** rephrase as the reviewer suggested.

10) Line 164: The potential ET during that day. **Comment:** How is the potential ET calculated here? Reference ET, potential ET, maximum ET are different concepts.

**Response:** Potential ET here was calculated through Penman-Monteith combination equation using hourly environmental data during the period from 1 April to 30 September (Fig. 3). This information has been mentioned in section 2.3.3 of the earlier version of manuscript, and further clarified in this revision.

11) Line 199-200: The upper boundary of the calculation was set as the atmospheric boundary condition, and the calculation involved actual precipitation, irrigation, and potential evapotranspiration rates for the crop cover. **Comment:** (1) how is the film mulching effects considered for the upper boundary condition? (2) how is the bare soil evaporation estimated as the upper boundary condition? (3) how is the upper boundary condition defined for the inter-cropping treatment? And the alternative mulching strips?

**Response: (1)** the film mulching effects on the upper boundary condition were modeled as proportionally damped such that $E_{p,a} = \beta \times E_p$, where $\beta$ is the area percentage without plastic film mulching in each experimental plot, and $E_p$ is potential ET estimated with the Penman-Monteith method. This issue has been clarified in the revision. **(2)** Basically, the bare soil evaporation ($E_a$) can be estimated via equation 6, which was provided in section 2.3.2. However, to be convenient in our coding, a simplified method proposed by Porporato *et al.* (2002) was employed to do this calculation, i.e., the evaporation was assumed to linearly increases with soil moisture ($\theta$) from 0 at the hygroscopic point ($\theta_h$), to $E_{p,a}$ at the field capacity ($\theta_{fc}$). For values of $\theta$ exceeding $\theta_{fc}$, evapotranspiration is decoupled from soil moisture and remains constant at $E_{p,a}$. We have added this information in the revision to clarify this point. **(3)** As already been mentioned in our response to question (1), we defined the upper boundary of alternative mulching strips according the ratio of plastic film mulching (i.e., 40%) and the potential ET estimated with the Penman-Monteith. However, we did not set specific upper boundaries for inter-cropping treatments, because the difference in surface soil evaporation between mono- and inter-cropping treatments could be relatively small when comparing with the transpiration in a growing season. We clarified this point in our revision and some potential uncertainties caused by this simplification also were include in this revision.

12) Line 226: In Table 4, **Comment:** Table 2?

**Response:** Sorry for the typo, it should be Table 2. We have corrected it in the revision.

13) Line 237: The profile averaged values of saturated drainage velocity ($K_s$) were 119, ˘ 129.36, 286.04, 189.42, 207.92, and 216.14 cm day$^{-1}$ at. . . **Comment:** It is not necessary to list these values in the text because they are already given in the table.

**Response:** The part has been reworded as suggested.

14) Line 269: . . .irrigation crop demand… **Comment:** Irrigation demand, crop demand are meaningful concepts in crop water requirement studies. What does the irrigation crop demand mean?

**Response:** Sorry for the misleading wording. It has been changed as "irrigation volume" in this revision.

15) Line 292: . . ., a slow-down or even a very light increase. . .**Comment:** A slow-down decrease or even light increase?

**Response:** Yes, it should be "A slow-down decrease or even light increase". Thanks for point it out.

16) Line 293-295: We checked all the soil moisture time series of NT1-NT6 during the entire growing season period (Fig.5), and no constant water content throughout the entire soil profile was detected in any of those selected plots, suggesting that our previous hypothesis that no steady-state flow took place during any irrigation events was supported. **Comment:** What is the purpose of this sentence? For any frequently irrigated soil profile, it is hard to reach a steady flow state.

**Response:** This sentence was used to prove that our previous hypothesis that no steady-state flow took place during any irrigation events was correct. We agree that it is a little bit redundant because it is hard to reach a steady flow state for any frequently irrigated soil profile. Following the reviewer's suggestion, it has been removed in the revision to solve the concern.

17) Line 298: . . .and strong potential evaporation may have hampered any effective infiltration from those precipitation events. **Comment:** It is the insufficient precipitation that attributes to the negligible infiltration rather than the strong evaporative demand.

**Response:** Thanks for pointing this out. We had corrected it in the revision.

18) Line 310: . . .and increased gradually as LAI became greater with crop development, . . . **Comment:** LAI has never been mentioned previously in the paper although it is very important information supporting discussions in the later sections.

**Response:** Since we don't have detailed information on LAI in this paper, we change the word "LAI" as "vegetation coverage" in the revision.

19) Line 315-316: The relative facility with which an excess of water in the soil was produced caused an important deep percolation, which became greater as it progressed further up the irrigation gradient. **Comment:** This sentence should be rephrased. It is confusing.

**Response:** It has been rephrased as: "The excess of water in the soil produced an important deep percolation, which became greater as the increasing of the irrigation quota."

20) Line 340: . . ., the soil moisture data-driven method. . .**Comment:** The soil moisture data-based method, might be a better description to this work.

**Response:** Corrected it in the revision as suggested.

21) Line 341: . . .the best. . . **Comment:** "the best" among which and which?

**Response:** "The best" has been replaced with "the better" here in the revision, and thus the revised sentence will be "Compared with the methods used in the literatures listed in Table 4, the soil moisture data-driven method used in this paper is more reliable because it produced the better fit between the numerical solution and the measured values of soil moisture content, even with vertical flow accounted for Guderle and Hildebrandt (2015)."

22) Line 344-345: . . ., which in turn suggested that both cropping systems and agronomic manipulation had limited influence on the accumulated ET during the growing season, . . .**Comment:** This is correct when preconditioned only.

**Response:** preconditions have been included here to solve the concern, and the statement in the revision has been reworded as "which in turn suggested that for the unmulched alfalfa and mulched maize, both cropping systems and agronomic manipulation had limited influence on the accumulated ET during the growing season".

23) Line 365: 4.2 Other estimated SWBCs. **Comment:** Does it mean the other SWBCs in this study or the SWBCs given by other people in the literatures?

**Response:** We mean the other SWBCs given in this study. It has been reworded as "the other SWBCs in this study".

24) Line 401: 4.3 Long-term effects on soil water budgets. **Comment:** Does this manuscript involve any long-term issues, either the parameters or the water balance budgets?

**Response:** Yes, this manuscript does involve some related issues of long-term management, i.e., the plots were designed to do long-term agronomic manipulation experiments (~10 years). Although the calculation was not based on long-term measurements, the long-term effects of agronomic manipulation on the soil hydrophysical properties and thus in turn on the soil water budget balances were analyzed. To solve the concern, this sentence has been changed as "section 4.3 Long-term effects on soil hydrophysical properties".

**References:**

Gardner W, Mayhugh M. 1958. Solutions and Tests of the Diffusion Equation for the Movement of Water in Soil 1. Soil Science Society of America Journal, **22**: 197-201.

Guderle M, Hildebrandt A. 2015. Using measured soil water contents to estimate evapotranspiration and root water uptake profiles - a comparative study. Hydrology & Earth System Sciences, **19**: 409-425.

Hanks RJ, Bowers SA. 1962. Numerical Solution of the Moisture Flow Equation for Infiltration into Layered Soils1. Soil Science Society of America Journal, **26**: 530.

Kang S, Zhang L, Liang Y, Hu X, Cai H, Gu B. 2002. Effects of limited irrigation on yield and water use efficiency of winter wheat in the Loess Plateau of China. Agricultural Water Management, **55**: 203-216.

Liu WZ, Hunsaker DJ, Li YS, Xie XQ, Wall GW. 2002. Interrelations of yield, evapotranspiration, and water use efficiency from marginal analysis of water production functions. Agricultural Water Management, **56**: 143-151.

Porporato A, D'Odorico P, Laio F, Ridolfi L, Rodriguez-Iturbe I. 2002. Ecohydrology of water-controlled ecosystems. Advances in Water Resources, **25**: 1335-1348.

Ross PJ. 2003. Modeling Soil Water and Solute Transport—Fast, Simplified Numerical Solutions. Agronomy Journal, **95**: 1352-1361.

Zhang Y, Kendy E, Qiang Y, Changming L, Yanjun S, Hongyong S. 2004. Effect of soil water deficit on evapotranspiration, crop yield, and water use efficiency in the North China Plain. Agricultural Water Management, **64**: 107-122.

Zuo, Qiang, Zhang, Renduo. 2002. Estimating root-water-uptake using an inverse method Soil Science, **167**: 561-571.

---

## Author Comment (AC4) · 29 Dec 2018

**Response to Dr. L. Brocca (SC1)**

Thank you very much for your nice comments, and sharing your related researches with us, which is found to be enlightening in revising our paper. Please find our point by point responses as below.

**Short comments**

1) While I mostly agree that in situ soil moisture data can be used for obtaining water fluxes, i.e., the soil water budget components, I also believe that uncertainties in the inversion approach are present.

**Response:** We totally agree that substantial uncertainties could be involved in the inverse approach, and this kind of uncertainties is even unavoidable in such calculation, irrespective of the used data model. For instance, uncertainties could occur due to errors in measurements of both the soil hydraulic parameters and soil moisture, or it may be due to the incomplete knowledge of boundary conditions and the limited volume that each set of soil moisture probes can monitor. To solve the concern, we will add a new section (**4.5 Uncertainty analysis**) to discuss all the possible uncertainties that could affect our results, and we argue that although such uncertainties inherently existed, the results are acceptable given the uncertainties are carefully considered and addressed.

2) The main issue I found is exactly related to the uncertainties in the inversion procedure. For instance, the six soil moisture probes show significant differences in the amount of irrigation for different probes; irrigation at NT1 is nearly double (1.8 times) of NT3. I do not understand how it might be possible as I believe that the amount of irrigation applied in the field is the same for all probes. Am I wrong? Indeed, the irrigation estimates strongly depend on the assumption behind equation (2) and, specifically, in the estimation of $S_{max.}$

**Response:** As also mentioned in our response to the last question proposed by Dr. Brocca, that kind of uncertainties have been thoroughly discussed and evaluated in the light of the effects on accuracy of our calculation (please see the newly added section of **4.5 Uncertainty analysis** in the coming revision). Upon the significant differences in the amount of irrigation for different plots noticed by Dr. Brocca, I would say you are right, but this was a misunderstanding probably due to our unclear wording in abstract. Indeed, the reason for NT1 got much higher irrigation than the other plots is that there was no plastic film mulching at this plot, while other plots (NT2-NT6) have (the cover percentage is about 40% of the total area). To solve the concern, we reworded and clarified this point in the revision. We agree that the irrigation estimates strongly depend on the assumption behind equation (2) and, specifically, in the estimation of $S_{max}$, and potential uncertainties caused by it and the reasonability behind it will be analyzed in the **4.5 Uncertainty analysis** in the revision.

3) Why is the comparison with in situ observations of irrigation not done? I believe it is needed definitely.

**Response:** We do have the in-situ observations of irrigation at field scale, but unfortunately not at plot scales, and the available in-situ observations of irrigation were also not directly measured through water meters, but instead through an indirect method. We used the power consumption of the pumping irrigation well ($P$) to obtain the actual irrigation amount of the plots ($Q$) through a well-built relationship, $Q = P \times \eta$, where $\eta$ is the ratio of the power consumption per unit water pumped

specifically determined at the field station. While we can calculate the more detailed irrigation data at plot scale with the recorded the irrigating time span for each plot, we believe the getting result is not accurate enough due to the potential inconsistence of water flow rate per unit time at this scale. That is why a compromising way was adopted in this paper, in which the estimated irrigation volumes of the six plots (through soil moisture data-based method) were averaged and tested against the observations (actual irrigation calculated from the power consumption) at field scale. Although the estimated average irrigation volume within the plots (831.6 mm) compares well with the actual irrigation volume (868.8 mm) determined through power consumption, we are aware that this is a drawback of our work, so that related discussions upon the possible uncertainties caused by it have been included in the earlier version of the manuscript and further evolved in this revision.

---

## Author Comment (AC5) · 2 Jan 2019

**Response to the anonymous reviewer 2# (RC2)**

**General comments**

ET is very important in water resource management. This manuscript presented an experimental study combing with soil water balance modeling and mathematical modeling. The research is worthy, well-structured and written. But I do not recommend to publish it in present form.

**Response**: We would like to thank the reviewer 2# for his accurate and frank review and used their precious suggestions to improve the paper. We tried to answer to all the comments made and we are ready to prepare and submit a new version of the manuscript. The point by point answers are written in blue.

**Specific comments**

1) The irrigation amount is the most important component in this study and needs to accurately measured. Unfortunately, the manuscript did not provide reliable information on it.

**Response**: We do have in-situ observations of irrigation at field scale, but unfortunately not at plot scales, and the available in-situ observations of irrigation were also not directly measured through water meters, but instead through an indirect method. While we can infer the more detailed irrigation data at plot scales with the recorded the irrigating time span for each plot, we are afraid that the derived data is not accurate enough due to the potential inconsistence of water flow rate per unit time at this scale. Given the irrigation volumes for the entire plots can be easily and reliably calculated through a well-built relationship between the power consumption of the pumping irrigation well ($P$) and the actual total irrigation amount of all plots ($Q$): $Q = P \times \eta$, (where $\eta$ is the ratio of the power consumption per unit water pumped, which was specifically determined at the field station), a compromising way was used to test the estimated irrigation, in which, the estimated irrigation volumes of the six plots were averaged and tested against the observations at field scale. Although the estimated average irrigation crop demand within the plots (831.6 mm, via the soil moisture data-based method) compares well with the actual irrigation volume (868.8 mm, via the relationship between $Q$ and $P$), we are still aware that without more detailed irrigation data for each plot is a drawback of the validation of the methods, so that related discussions upon the possible uncertainties caused by it will be included in the earlier version of the manuscript and further evolved in this revision.

2) two irrigation amounts were used in this study, one calculated from the difference between $S_{max}$ and $S_{ini}$, the other estimated from power consumption. But I do (not) understand which is the real value and which is the measured value (L272).

**Response:** Sorry for the misleading wording in the manuscript. We considered the irrigation amounts determined from power consumption is the "real value", and the one calculated from the difference between $S_{max}$ and $S_{ini}$ as the "estimated value" or the value to be validated. We will reorganize this part to clarify it.

3) The definition of $S_{max}$ was not clearly given.

**Response**: Clear definition will be included in the coming revision, i.e., "Where $S_{max}$ is the maximum soil water storage of root zone (0-110cm) after one irrigation event began".

4) According to the text, the irrigation was delivered at a rate of 2250/ha/min (L264) and each irrigation event lasted 20-30 min, therefore the irrigation amount was about 6 mm for each time and total irrigation amount was about 60 mm for the growing season. If it is not mistake, it is much lower than the estimation and is not acceptable.

**Response:** Thanks for point out this careless mistake arising from our calculation, this value should be ~32500 L/ha/min or 3.25 mm/min in the manuscript. We have corrected this mistake in our revision.

5) I think it is possible to make accurate measures with water meter in such small experimental plot.

**Response:** Yes, it is reasonable but failed to collect accurate measures with water meter in the small experimental plots, and we are planning to install this kind of instruments before the start of the coming growing season. We are aware that without such detailed irrigation data for each plot is a drawback of the validation of the methods, so that related discussions upon the possible uncertainties caused by it will be included in the earlier version of the manuscript and further evolved in this revision. Please also see our response to Question 1 of RC2.

6) About the mathematical model, the setting is quite important. In section 2.3.2, equation (7) shows that the lower boundary was defined by soil matric potential (L175), which in section 2.3 3), it was set to free drainage (L202). It is not acceptable.

**Response:** While we don't think there are any contradictions between the setting of the lower boundary as free drainage and the equation (7) in which it was defined by soil matric potential, the 1-D Richards Equation (equation 4-7) will be reorganized to eliminate the potential confusion.

7) Moreover, root uptake is also an important factor in water redistribution simulation, but the manuscript did not give the information about the root distributions of the crops.

**Response:** The reason for the omission of root distribution information in the manuscript is that the inverse model we adopted does not require any a priori information of root distribution parameters. Root water uptake parameters were estimated by minimizing the residuals between simulated and measured soil water contents (Zuo *et al.*, 2002; Guderle and Hildebrandt, 2015). That is, the distribution of soil water contains the information of root-water-uptake distribution. It's a great advantage of this method because the parameters of those rooting profile functions are cumbersome to measure in the field, and the relevance for root water uptake distribution is also uncertain. This issue has been discussed in previous work of Schneider *et al.* (2010); Guderle and Hildebrandt (2015), and some related discussions also be included in the revision. To solve the concern, general root distribution information of maize, alfalfa and pea in this region has been included in section 2.2 (Site description) in the revision.

8) About the location of the TDR systems in the experimental plots. Because the irrigation method is furrow irrigation, the soil water contents are different with location. From table 2, the saturate water content is about 34% and the field capacity is about 20%. If the preferential flow is limited, as suggested in the manuscript, the water content should be higher than 20%, even 24h after the irrigation event. But the measured value is only 21.9% for $S_{max}$ and 14% for S24. Therefore, I doubt the TDR systems were not in the suitable location. Therefore, the information is not enough and correct.

**Response:** Thanks for pointing out this limitation. We agree that soil water contents following furrow irrigation may be different at various locations in a ridge film-mulched field, however, we also argue here that this kind of different can be largely neglected in practice. The reasons are as followed: 1) while TDR probes in our field experiment were installed under the film-mulched ridges, the height of ridge shoulders in the experimental plots is relatively low (<5cm), substantial infiltration could occur through the film holes made for maize-growth (see Fig.1 in the appendices); 2) lateral water transfers could be substantially enhanced during the period of irrigation due to the soil water potential differences between ridges and furrows. This judgement also can be supported by some researchers conducted at similar environments, i.e., Zhang *et al.* (2016). Given the effect of plastic mulched furrow irrigation on soil water distribution remains elusive (Abbasi *et al.*, 2004; Zhang *et al.*, 2016), potential uncertainties caused by the location of TDR install are discussed to solve the reviewer's concern in the revision.

The reviewer noticed that "the measured value is only 21.9% for $S_{max}$ and 14% for $S_{24}$", and argued that "the water content should be higher than 20%, even 24h after the irrigation event". As such, it was doubt by the reviewer that the preferential flow may occur in the soil profiles and the TDR systems were probably not in the suitable location. After carefully checking our data, we now can confidently say that our judgement ("preferential flow is limited") is reasonable. **Firstly**, both the $S_{max}$ and $S_{24}$ were defined as the moisture storages within the entire soil profiles in this work, so they are the average values of soil moisture at different stages of irrigation, which can never be kept at a relatively high level (i.e., $> s_{fc}$, the soil field capacity), due to the relatively large hydraulic conductivities of coarse-textured sandy soil. I guess the reviewer misunderstood the conceptions of soil water content and the soil water storage we provided in the manuscript, probably due to our unclear wording. We will clarify the descriptions in our revision. **Secondly**, if the TDR systems were not in the suitable locations, different situations should happen at NT1 (unmulched, and with almost flat surface) and other plots (NT2-6, mulched). However, the fact is that almost similar results were observed in the soil moisture dynamics at all the soil profiles following an irrigation event (see Fig.2 in the appendices).

**Figures.**

[Figure]

**Figure 1. The furrow and ridge in the experimental plots**

[Figure]

**Figure 2. Soil water dynamics of unmulched (NT1) and mulched plots (NT2-4) in different soil depth after one irrigation event.** *The biggest difference of soil moisture dynamics between unmulched flat plot (NT1, which is independent of TDR location) and film-mulched ridge plot (NT2-6, which is affected by TDR location) was appear in the top-20cm soil layer (blue line). With the increase of soil depth, this difference is fade away, the soil moisture curve of 100cm (green line) is very close between NT1 and other plots.*

**References:**

Abbasi F, Feyen J, Genuchten MTV. 2004. Two-dimensional simulation of water flow and solute transport below furrows: model calibration and validation. Journal of Hydrology, **290**: 63-79.

Guderle M, Hildebrandt A. 2015. Using measured soil water contents to estimate evapotranspiration and root water uptake profiles – a comparative study. Hydrology and Earth System Sciences, **19**: 409-425.

Schneider CL, Attinger S, Delfs JO, Hildebrandt A. 2010. Implementing small scale processes at the soil-plant interface - the role of root architectures for calculating root water uptake profiles. Hydrology and Earth System Sciences,14,2(2010-02-12), **14**: 279-289.

Zhang YY, Wu PT, Zhao XN, Zhao WZ. 2016. Measuring and modeling two-dimensional irrigation infiltration under film-mulched furrows. Sciences in Cold & Arid Regions, **8**: 419-431.

Zuo, Qiang, Zhang, Renduo. 2002. Estimating root-water-uptake using an inverse method. Soil Sci, **167**: 561-571.